# Chromosome-scale genome assembly reveals how repeat elements shape non-coding RNA landscapes active during newt limb regeneration

## Graphical abstract

## Authors

Thomas Brown, Ketan Mishra, Ahmed Elewa, ..., Nicholas D. Leigh, Maximina H. Yun, András Simon

## Correspondence

nicholas.leigh@med.lu.se (N.D.L.), maximina.yun@tu-dresden.de (M.H.Y.), andras.simon@ki.se (A.S.)

## In brief

Understanding molecular mechanisms behind newts' exceptional regenerative capability has been limited because of their large, repeat-element-rich genome. Brown et al. present the most complete chromosome-scale assembly of any giant genome to date. They reveal that repeat elements are expressed during regeneration and shape the non-coding RNA genomic landscape.

## Highlights

- Chromosome-scale assembly with the highest contiguity among giant genomes

- Regulation of hAT transposons and circRNAs during limb regeneration

- Annotation of miRNAs identifying novel species-specific miRNAs

- Repeat element expansion associated with miRNA distribution and circRNA formation

 Brown et al., 2025, Cell Genomics 5, 100761
February 12, 2025 © 2025 The Author(s). Published by Elsevier Inc.

CellPress

## Article

# Chromosome-scale genome assembly reveals how repeat elements shape non-coding RNA landscapes active during newt limb regeneration

Thomas Brown,[1,2,16] Ketan Mishra,[3,16] Ahmed Elewa,[4,16] Svetlana Iarovenko,[5,16] Elaiyaraja Subramanian,[3]
Alberto Joven Araus,[3] Andreas Petzold,[1] Bastian Fromm,[6] Marc R. Friedländer,[7] Lennart Rikk,[8] Miyuki Suzuki,[9]
Ken-ichi T. Suzuki,[10] Toshinori Hayashi,[11,12] Atsushi Toyoda,[13] Catarina R. Oliveira,[5] Ekaterina Osipova,[14]
Nicholas D. Leigh,[8,16,*] Maximina H. Yun,[2,5,15,16,*] and András Simon[3,16,17,*]

[1]DRESDEN-concept Genome Center (DcGC), Center for Molecular and Cellular Bioengineering, Technische Universität Dresden, 01307 Dresden, Germany
[2]Max Planck Institute of Molecular Cell Biology and Genetics, 01307 Dresden, Germany
[3]Department of Cell and Molecular Biology, Karolinska Institute, 171 65 Stockholm, Sweden
[4]Department of Biology, Augsburg University, Minneapolis, MN 55454, USA
[5]CRTD Center for Regenerative Therapies Dresden, Technische Universität Dresden, 01307 Dresden, Germany
[6]The Arctic University Museum of Norway, UiT – The Arctic University of Norway, 9006 Tromsø, Norway
[7]Science for Life Laboratory, Department of Molecular Biosciences, The Wenner-Gren Institute, Stockholm University, 114 18 Stockholm, Sweden
[8]Molecular Medicine and Gene Therapy, Wallenberg Centre for Molecular Medicine, Lund Stem Cell Center, Lund University, 221 84 Lund, Sweden
[9]Division of Biology and Biological Engineering, California Institute of Technology, Pasadena, CA 91125, USA
[10]Emerging Model Organisms Facility, Trans-scale Biology Center, National Institute for Basic Biology, Okazaki, Aichi 444-8585, Japan
[11]Program of Biomedical Science, Graduate School of Integrated Sciences for Life, Hiroshima University, Higashi-Hiroshima, Hiroshima 739-8511, Japan
[12]Amphibian Research Center, Hiroshima University, Higashi-Hiroshima, Hiroshima 739-8511, Japan
[13]Comparative Genomics Laboratory, Department of Genomics and Evolutionary Biology, National Institute of Genetics, Mishima, Shizuoka 411-0801, Japan
[14]LOEWE Centre for Translational Biodiversity Genomics, Senckenberganlage 25, 60325 Frankfurt, Germany
[15]Physics of Life Excellence Cluster Dresden, 01307 Dresden, Germany
[16]These authors contributed equally
[17]Lead contact
*Correspondence: nicholas.leigh@med.lu.se (N.D.L.), maximina.yun@tu-dresden.de (M.H.Y.), andras.simon@ki.se (A.S.)

## SUMMARY

Newts have large genomes harboring many repeat elements. How these elements shape the genome and relate to newts' unique regeneration ability remains unknown. We present here the chromosome-scale assembly of the 20.3 Gb genome of the Iberian ribbed newt, *Pleurodeles waltl*, with a hitherto unprecedented contiguity and completeness among giant genomes. Utilizing this assembly, we demonstrate conserved synteny as well as genetic rearrangements, such as in the major histocompatibility complex locus. We provide evidence suggesting that intronic repeat elements drive newt-specific circular RNA (circRNA) biogenesis and show their regeneration-specific expression. We also present a comprehensive in-depth annotation and chromosomal mapping of microRNAs, highlighting genomic expansion profiles as well as a distinct regulatory pattern in the regenerating limb. These data reveal links between repeat elements, non-coding RNAs, and adult regeneration and provide key resources for addressing developmental, regenerative, and evolutionary principles.

## INTRODUCTION

Repeat elements constitute the bulk of eukaryotic genomes, spanning ~50% in human to ~67% in lungfish.[1] They are often implicated in the evolution of genome size, influencing gene expression, enriching the transcriptome, and controlling cell properties such as stemness.[2–5] How the genome is shaped by repeat elements, and how these elements are related to regeneration, has been difficult to discern.

The Iberian ribbed newt *Pleurodeles waltl* boasts an impressive regenerative roster, capable of rebuilding lost limbs and regenerating damaged tissues in complex organs, including brain, heart, and eye.[6–10] Use of this model species would be greatly enhanced by a high-quality genome assembly and annotation.

This has, however, been a challenge, primarily due to the large genome size of ~20 Gb and considerable enrichment in repeat element sequences.[11,12]

Here, we have employed long-read sequencing and chromosome conformation capture techniques to generate a highly contiguous and complete chromosome-scale assembly of the *P. waltl* genome. In addition, we characterized the repeat element landscape, demonstrating that it makes up 74% of the genome. Owing to the large proportion of repeats, we focused on non-coding RNAs, circular RNAs (circRNAs) and microRNAs (miRNAs), whose origins have been linked to transposable elements.[13,14] CircRNAs were initially considered a by-product of splicing, but recent studies have uncovered several functions, such as in transcription regulation,[15] intermolecular interactions,[16] and regulation of stemness.[17] The formation of circRNAs is facilitated by long introns and flanking repeat elements, both of which are characteristics of the salamander genome.[18,19] However, the extent to which these genomic features impact circRNAs in salamanders remains unexplored. Utilizing this new genome assembly, we annotated circRNAs and demonstrated their association with hobo-Ac-Tam3 (hAT) repeat elements. Similar to circRNAs, miRNAs that originated from repeat element transposition events in both plants and mammals have been identified.[20,21] We establish here a chromosome-level miRNA annotation and show that many of them are hosted in long terminal repeat (LTR) elements.

Collectively, our findings support a role for repeat elements in shaping the genome of the highly regenerative *P. waltl* and enabling species-specific circRNA formation and miRNA expansion.

## RESULTS

### Sequencing, chromosome-scale assembly, and annotation of the *P. waltl* genome

To overcome the challenges inherent in sequencing and assembling a giant genome, we took advantage of the highly accurate Pacific Biosciences (PacBio) HiFi sequencing technology (Figure 1A). We sequenced genomic DNA from a female (ZW) *P. waltl* (Figure S1), generating 44,683,779 long reads, representing 41× genome coverage. We used *HiFiasm*[22] and purge-dups[23] and created a 20.3 Gb primary contig assembly with an N50 of 45.6 Mb and N90 of 11.1 Mb (Figure 1B). In addition, we produced two haplotype-phased assemblies, which had total sizes of 19.8 and 19.9 Gb with contig N50s of 20.5 and 20.7 Mb and contig N90s of 5.1 and 5.1 Mb, respectively. The size of the genome we determined is in agreement with previous estimates.[11] The contiguity of this *P. waltl* genome is increased by an order of magnitude compared to the lungfish *Protopterus annectens* and the Montseny brook newt *Calotriton anoldi* assemblies,[24,25] two orders of magnitude compared to the axolotl *Ambystoma mexicanum* assembly,[26] and four orders of magnitude compared to the previous, short-read-based *P. waltl* assembly[11] (Figure 1C). k-mer-based analysis indicated a level of heterozygosity of 0.39% (Data S1). The assemblies are also highly complete, with BUSCO completeness of 97.2% for the primary assembly and 96% and 96.5% for the two haplotype-phased assemblies (Figure 1B and Table S1).

Next, we set out to generate a high-resolution chromosome-scale assembly based on chromosome conformation capture (Hi-C) Illumina reads. We generated two Arima Hi-C libraries from heart tissue (6 billion read pairs—1.8 Tbp) of the same individual sequenced above (Figure S1). Following scaffolding with *salsa2*[27] and manual curation (Figures 1A and S2), we obtained a chromosome-level primary assembly with a scaffold N50 of 1.24 Gb in which 99.6% of the contigs could be assigned to the 12 chromosomes (Figure 1B). We also obtained chromosome-level assemblies for the two haplotype assemblies, with scaffold N50s of 1.36 and 1.2 Gb and 98.9% and 99.1% of each assembly assigned to the 12 chromosomes (Table S1). Each assembly of chromosomes 1–4 were split into two parts due to technical limitations in processing scaffolds larger than 2 Gb. The percentage of chromosome incorporation of contigs is on a par with the highest among all reported giant genomes at 99.6%.[24] The number and length of the chromosomes agree with the previously reported *P. waltl* karyotype.[11] At the structural level, Hi-C contact analysis (Figure 1D) and subsequent haplotype-phased assemblies (Figure S3) indicate the existence of two inversions between the two haplotypes of the sequenced individual, one in the central region of chromosome 2 (Figure S4 and Data S2) and another one on chromosome 5 (Figure S5 and Data S3).

To facilitate the annotation of genomic loci, we took advantage of the PacBio Iso-seq platform and generated full-length mRNA sequences for brain (1 Gb[28]) and spleen (0.86 Gb) of the same newt used for DNA sequencing. These were combined with PacBio Iso-seq data from adult limb blastema (1.76 Gb), *de novo* transcriptomes,[11,29] and Augustus predictions. Based on BUSCO analysis, we predicted a total of 18,799 conserved protein coding genes comprising 96.7% of vertebrate single-copy orthologs, with only 2.6% missing (Figure 1B), confirming the high completeness of the *P. waltl* assembly in comparison to other genomes (Figure 1E). Further, we found that *P. waltl* had a similar number of protein-coding genes compared to other vertebrates (Figure 1E), indicating that whole-genome duplication events are not responsible for the expansion of the *P. waltl* genome.

### Synteny conservation

To investigate syntenic relationships, we first examined the arrangement of 17 ancestral chordate linkage groups (CLGs).[30] This analysis revealed an enriched localization of CLGs within 43 blocks across the chromosomes of *P. waltl* (Figures 2A and S6 and Table S2). Given its phylogenetic position providing a bridge between teleosts and tetrapods,[31] we used the spotted gar (*Lepisosteus oculatus*) as a reference point together with the more closely related *A. mexicanum* for cross-species comparisons. Next, we generated a ribbon plot to visualize the syntenic blocks across the chromosomes of the three compared species (Figure 2B). The data identified both conservation and divergence at the macrosynteny level. As such, we found that the locations of some of the *L. oculatus* linkage groups (LGs) were conserved between *P. waltl* chromosome 1 and *A. mexicanum* chromosome 6, respectively, containing genes from LGs 2 and 4, as well as CLGs C, L, F, I, and Q. Similarly, LG 8 was found in chromosome 4 of *P. waltl* and chromosome

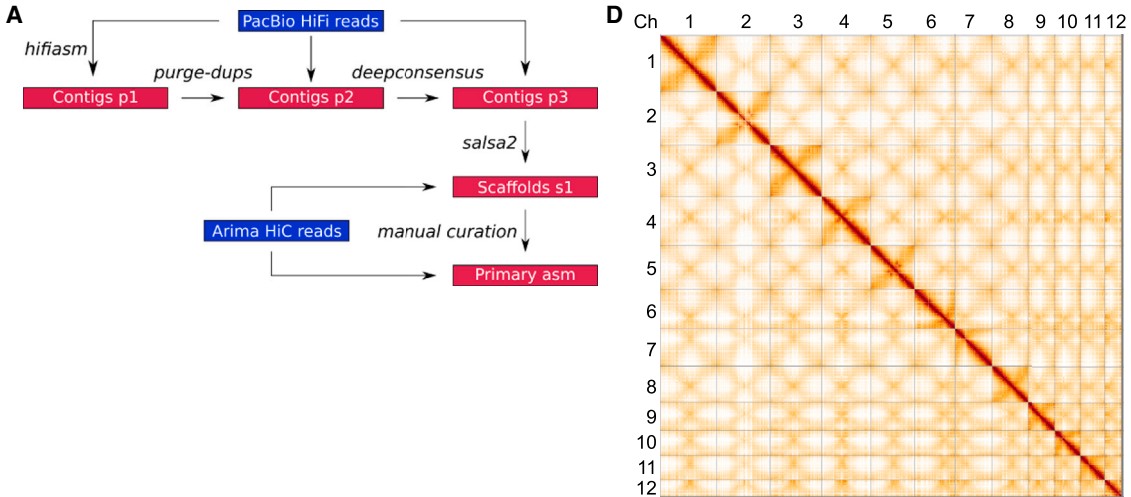

**A** Schematic representation of sequencing/assembly workflow

**D** Hi-C interaction heatmap

**B**

| P. waltl genome assembly | Length | Number | N50 | N90 | % in 12 Chr |
|---|---|---|---|---|---|
| Contigs | 20.3Gb | 1,186 | 45.6Mb | 11.1Mb | |
| Scaffolds | 20.3Gb | 265 | 1.24Gb | 913Mb | 99.6% |

| BUSCO | Complete | Single | Duplicated | Fragmented | Missing |
|---|---|---|---|---|---|
| Genome | 97.2% | 94.9% | 2.3% | 1.5% | 1.3% |
| Annotation | 96.7% | 41.0% | 55.7% | 0.7% | 2.6% |

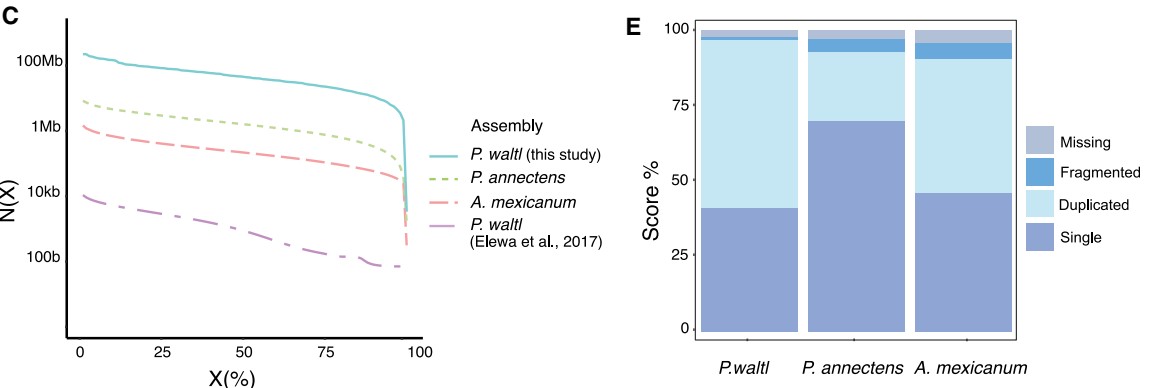

**C** Contig N(X) plot

Assembly:
- *P. waltl* (this study)
- *P. annectens*
- *A. mexicanum*
- *P. waltl* (Elewa et al., 2017)

**E** Gene completeness (BUSCO)

- Missing
- Fragmented
- Duplicated
- Single

*P. waltl*, *P. annectens*, *A. mexicanum*

**Figure 1. Sequencing and chromosome-level assembly of the *P. waltl* genome**

(A) Schematic representation of the sequencing and assembly strategy.

(B) *P. waltl* genome assembly features (top) and BUSCO assessment (bottom).

(C) Contig N(X) plot showing what percentage of each assembled genome (X) is contained within pieces at least N(X) bp in size. Shown are contig statistics from *P. waltl* (this study and Elewa et al.[11]), *Protopterus annectens*,[24] and *Ambystoma mexicanum*.[26]

(D) Hi-C interaction heatmap of contact data for scaffolded genome. Individual scaffolds are delineated. Denser areas of red signal off diagonal represent interactions between the arms of the same chromosome.

(E) Gene completeness based on BUSCO single-copy Vertebrata orthologs (*n* = 3,354). Scores are based on annotations for *P. waltl*, *P. annectens*, and *A. mexicanum*.

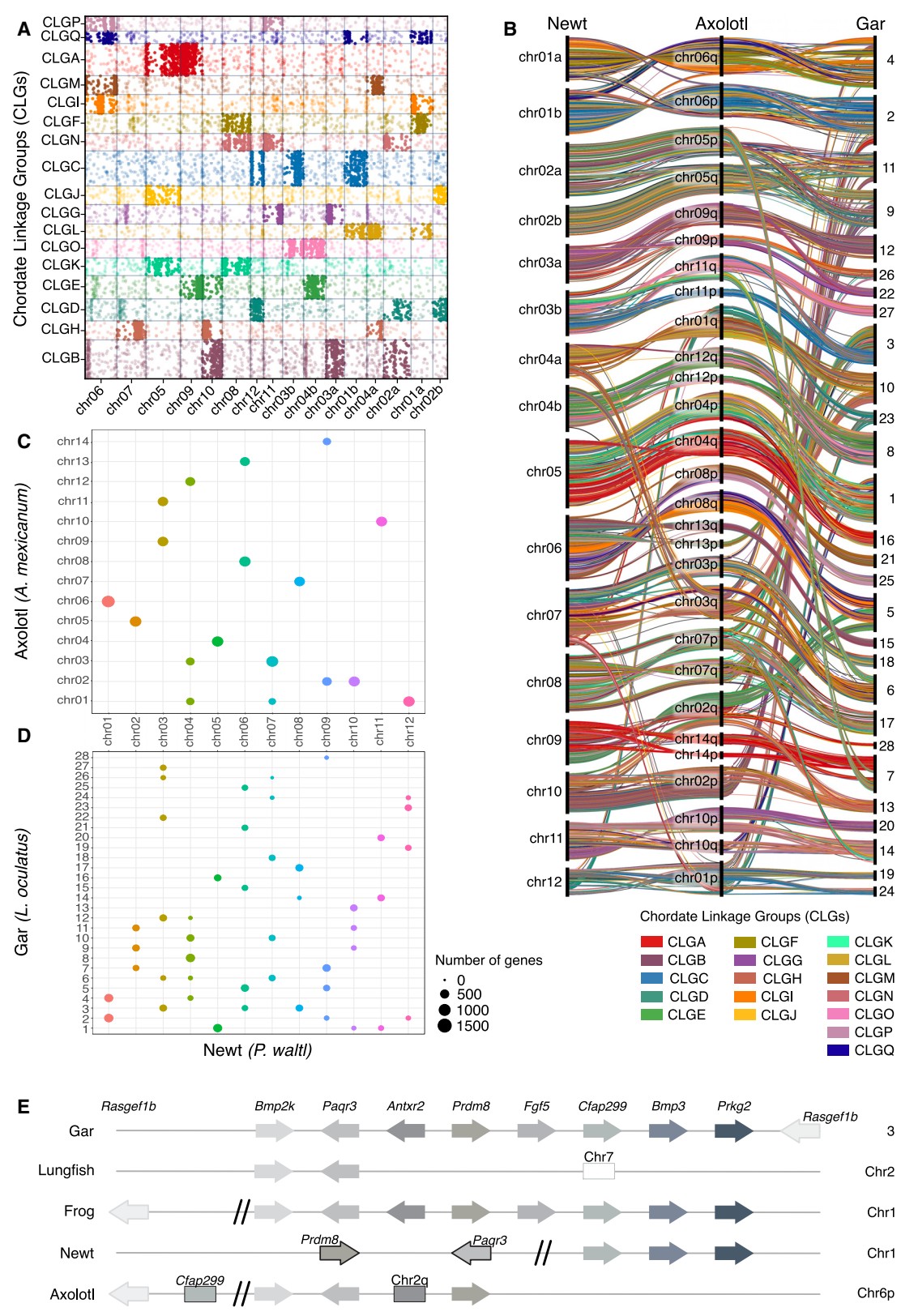

*(legend on next page)*

12 of *A. mexicanum*. In contrast, LG 10 was split between chromosomes 4 and 7 in *P. waltl* and chromosomes 1 and 3 in *A. mexicanum* (Figures 2B and S6 and Tables S3 and S4). Furthermore, we observed many genes within smaller units of syntenic blocks among salamander genomes compared to gar (Figures 2C and 2D and Tables S5 and S6).

Next, we compared 1:1 orthologous genes between the three species. Our findings, based on 6,434 1:1 orthologs between *P. waltl* and *A. mexicanum* (Figure S6D) and 11,925 between *P. waltl* and *L. oculatus* (Figure S6E), indicate a higher level of linear synteny between the two salamander species than between the newt and the evolutionarily more distant gar. We also observed macrosyntenic correspondence to other organisms such as *Xenopus* (Figure S7), lungfish (Figure S8), and humans (Figure S9). Of note, the analysis of syntenic boundaries between *P. waltl*, axolotl, and *Xenopus* provided independent evidence for the aforementioned inversion of chromosome 5 in one haplotype, as indicated by the Hi-C data (Figures S5 and S7). These data show substantial macrosyntenic conservation of several ancestral CLGs in *P. waltl*.

To analyze microsynteny, we first examined the locus encoding the *Tig1* gene, which encodes a determinant of positional identity of cells during limb regeneration.[32] We observed a conserved organization across representative vertebrate genomes, with *Schip1* and *Mfsd1* located at the 5′ end and *Gfm1*, *Mlf1*, and *Shox2* at the 3′ end of the *Tig1* locus (Figure S10).

Next, we focused on gene loss in *P. waltl* to explore evolutionary changes in genome organization. Notably, *Prdm9*, a gene recurrently lost throughout evolution,[33] is absent from the *P. waltl* genome (see STAR Methods). In addition, we identified the loss of *Fgf5* in salamanders. The loss of *Fgf5* in *P. waltl* coincides with a large-scale chromosomal rearrangement, evidenced by a 58 Mb gap between *Prdm8* and *Cfap299* (Figure 2E). Comparing microsynteny of the *Fgf5* locus across species, we observed that gar and frog both have a preserved *Fgf5* gene locus. This suggests that the disruptions present in lungfish and salamander genomes were independent events that occurred at their respective loci. Interestingly, the two salamander species also showed variations between each other, with *Rasgef1b*, *Bmp2k*, and *Antxr2* missing in newts but present in axolotls (Figure 2E). The evolutionary dynamics of this locus suggests active selection against these genes among vertebrates.

To explore a well-known evolutionarily dynamic locus, we examined the major histocompatibility complex (MHC), which is expanded in the axolotl genome.[26] Our analysis revealed a bimodal distribution of genes at the MHC locus on chromosome 6 in *P. waltl*: one region spans from *Trim39* to *Gabr1*, followed by a large region lacking MHC-related genes and a second region spanning from *Tubb* to *Gln1* (Figure S11). To better understand this bimodal distribution, we compared the *P. waltl* protein-coding genes from *Trim39* to *Gln1*, including non-MHC genes, to the corresponding regions in the axolotl genome. The MHC gene region on *P. waltl* chromosome 6 matched the axolotl MHC locus on chromosome 13, while the large non-MHC region showed high homology to axolotl chromosome 8. These observations corroborate our macrosynteny analysis, which showed that newt chromosome 6 shares conserved chromosomal linkages on chromosomes 8 and 13 of axolotl and LGs 5, 15, 21, and 25 in gar (Figure 2B). In addition, we observed a similar pattern with newt chromosome 9, it being split between axolotl chromosomes 2 and 14 (Figure 2B). These findings suggest that *P. waltl* underwent chromosomal rearrangements potentially involving fusion events, which may account for the two fewer chromosomes in *P. waltl* compared to *A. mexicanum*.

Altogether, we utilized the highly contiguous chromosome-scale assembly of *P. waltl* to provide insights into both macro- and microsyntenic conservation.

## Transposable elements underlie *P. waltl* genome expansion

Proliferation of transposable elements (TE) has emerged as a driving force behind the expansion of giant genomes, as observed in lungfish and axolotl.[1,24,34] Expansion of Gypsy retrotransposons and DNA Harbinger transposable elements was estimated to account for two-thirds of the repetitive content based on the short-read *P. waltl* assembly.[11] We performed repeat masking of the present genome assembly, uncovering that 74% of the genome (corresponding to 15 Gb) is made up of repeat elements (Figure 3A). Repeat content is higher than that of the giant axolotl genome based on our own (68%, Figure 3A) and previously reported analyses.[26] This may be attributable to either a higher accumulation of transposable elements in the *P. waltl* genome or the higher contiguity and completeness of its assembly. Further, the percentage of repeat element sequences in the *P. waltl* genome is comparable to that of lungfish,[1,24] which exhibits the highest reported repeat content found in the animal kingdom. Unlike the axolotl genome, in which LTR transposable elements are dominant (45% of repeats, 8.7 Gb), the DNA repeat class is the major contributor (51% of repeats,

**Figure 2. Comparative synteny of the *P. waltl* genome**

(A) Oxford plot depicting the locations of chordate linkage groups (CLGs). A total of 14,062 best-reciprocal-hit orthologs were identified between the *P. waltl* chromosomes and 17 ancestral CLGs. The dense rectangular blocks of dots represent units of deeply conserved synteny. Fisher's exact test (FET) was used to calculate the significance of interactions between each scaffold and linkage group. Dots are colored by CLG, with solid dots indicating the FET $p \leq 0.05$ and translucent dots depicting the FET $p > 0.05$.

(B) Ribbon plot showing syntenic blocks of the newt, axolotl and gar genomes. Best-reciprocal-hit orthologs are connected by ribbons between the *P. waltl*, the *A. mexicanum*, and the *L. oculatus* genomes and colored based on their identification as proteins from the 17 CLGs as in (A).

(C and D) Regions of the newt, axolotl, and gar genomes identified pairwise as containing co-localized blocks of genes with the CLGs between axolotl and newt (C) and gar and newt (D). The size of each circle corresponds to the number of genes in each identified syntenic block. Circles are colored by CLG. Individual numbers are also available in Tables S2–S6.

(E) Microsynteny analysis of the *Fgf5* locus between gar, lungfish, frog, newt, and axolotl. The newt locus has experienced various disruptions, including a 58 Mb gap between *Prdm8* and *Cfap299* as well as an inversion of order between *Paqr3* and *Prdm8*. Arrows represent the relative position and direction of genes on chromosomes, retrogenes are marked by white rectangles, and filled rectangles imply a different chromosomal position.

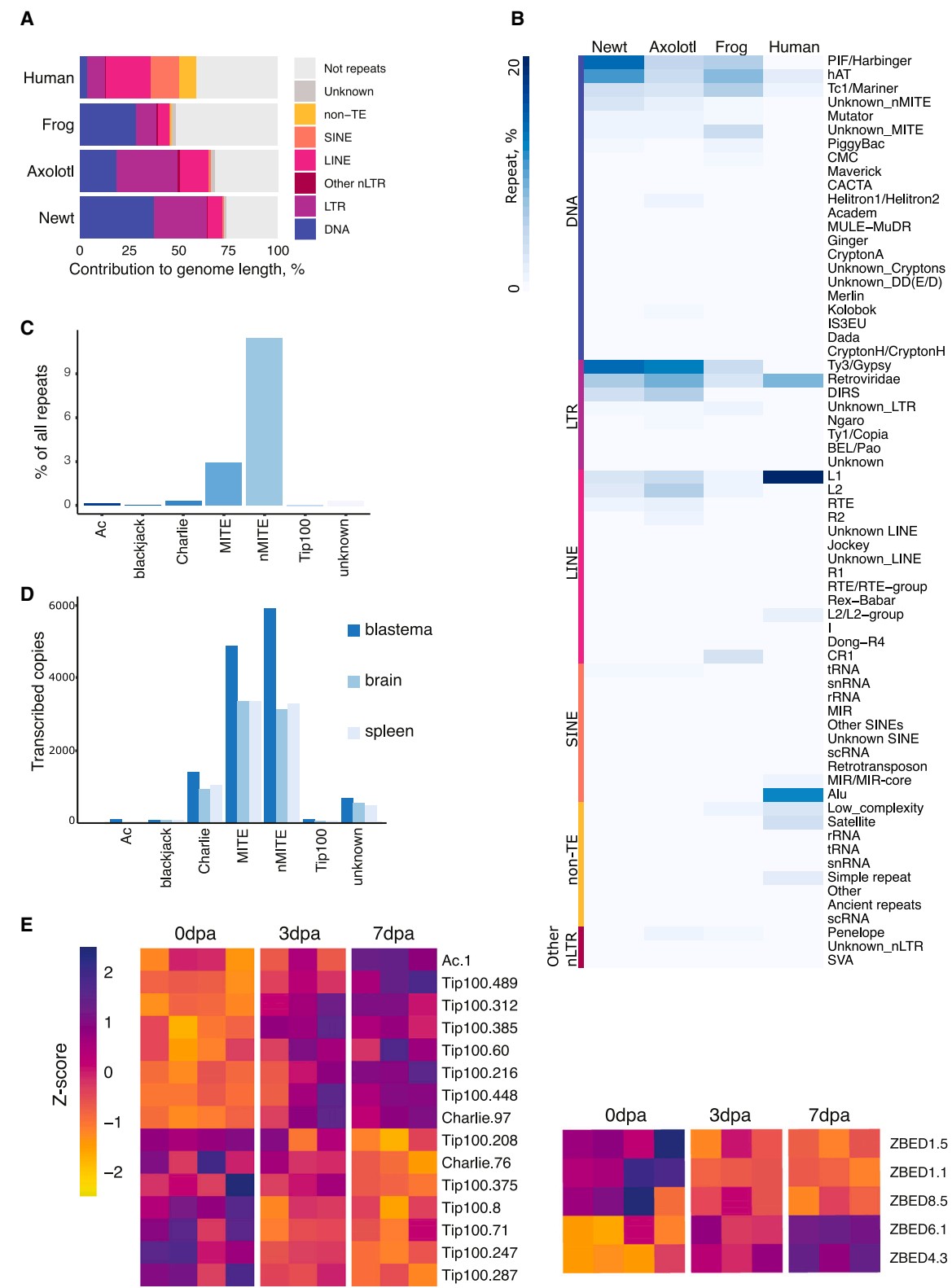

(legend on next page)

7.6 Gb) to the *P. waltl* genome (Figure 3A). In-depth analysis of the genomic contributions of each repeat superfamily (Figure 3B) revealed that Gypsy (19.7% of repeats, 3 Gb) and Harbinger (19.6% of repeats, 2.9 Gb) elements are among the most represented superfamilies of LTRs and DNA repeats, respectively, as previously suggested.[11] Notably, we also uncovered a large contribution of DNA/hAT transposable elements (15% of repeats, 2.3 Gb). Among the predominant DNA/hAT elements, the miniature inverted repeat transposable element (MITE) and non-autonomous MITE (nMITE) types make up 15% of all repeats in the assembly (Figure 3C).

To gain insights into the expansion history of transposable elements in the *P. waltl* genome, we performed Kimura distance analysis, whereby the number of substitutions to the consensus sequence of each element was used to estimate its relative age. These analyses highlighted differential expansion kinetics between the main transposable element types (Figures S12A and S12B), with DNA/hAT and Harbinger elements having undergone one wave of expansion with overlapping periods and LTR/Gypsy elements exhibiting a longer period of basal activity (Figure S12). Thus, the *P. waltl* genome has sustained several waves of transposable-element-driven expansion.

Bursts of transposition are key sculptors of genomes over time. We found that DNA/hAT elements were distributed within intergenic regions (77.69%), introns (22.26%), and exons (0.02%) (Figure S13). In fact, this distribution pattern characterized all other repeat elements (Figure S11). Transposable element contributions have resulted in an increase in intronic as well as intergenic dimensions that is several orders of magnitude larger than in frogs and humans[34] (Figure S14).

The hAT elements contribute significantly more to the expansion of the intergenic region (*p*adj < 1.47e−321, see STAR Methods) as opposed to the intronic length. Among DNA repeat elements, hATs have a significant expression in the early phases of limb regeneration in *P. waltl* (Figure S15). To further characterize the early expression of hAT elements, we mined the PacBio Iso-seq transcriptomes from adult regenerating limb blastema, brain, and spleen (Figure 3D). We found that all hAT types were expressed in these tissues, albeit to different degrees. Notably, these include several hATs longer than 2,000 bp (Figure S16). Furthermore, the number of expressed hAT elements was higher in the regenerating limb blastema compared to other tissues examined (Figures S16 and S17). Differential expression analysis uncovered dynamic transcriptional changes in several Tip100, Charlie, MITE, and Ac hATs during limb regeneration (Figure 3E and Data S4), both during wound healing (3 days post-amputation [dpa]) and at early bud blastema stages (7 dpa). Further, we found various zinc finger BED-type (ZBED) gene family members, which derive from hAT transposons following a "domestication" event,[35] that were upregulated during limb regeneration (Figure 3E and Data S5[11]) and also differentially expressed across several tissues, including the germline (Figure S18). We corroborated the RNA sequencing (RNA-seq) findings with quantitative reverse-transcription PCR (RT-qPCR) by focusing on the expression of two ZBED family members (*Zbed1* and *Zbed6*) and a hAT (*Charlie 76*) (Figure S19).

## Repeat elements shape the circRNA landscape

Backsplicing events lead to the formation of circRNAs, covalently closed circular structures that lack a 5′ cap and a 3′ poly(A) tail present on linear mRNAs.[36] As recent evidence found that active transposable elements facilitate circRNA genesis,[13,37] we set out to characterize circRNAs in *P. waltl.* We found and annotated circRNAs expressed across a range of tissues utilizing a previously published dataset of 18 total RNA-seq libraries[11] mapped against the present assembled genome (Figure S20).

For accurate prediction,[38,39] we employed three independent circRNA detection programs that all predict circRNAs from back-spliced junction reads: CIRCexplorer2,[40] Find_Circ2,[41] and CIRI2[42] (Figure S20). We found 9,799 circRNAs that were detected by all three programs, all of which annotated to known splice sites (Figure 4A and Table S7). In mammals, long introns flanking back-spliced junctions facilitate circularization of RNA.[36,43] In accordance with this, we observed a statistically significant increase in the mean length of introns flanking the predicted circRNAs compared to random permutations (*p* < 2.2e−16, Figures 4B and S21). Among genes that host circRNAs, we found 20 circRNA "hotspot" genes that harbored 15 or more different circRNA isoforms (Figure 4C and Table S8). Inverted-repeat elements in flanking introns are also associated with the biogenesis of circRNAs.[44] Analysis of this pattern in *P. waltl* revealed that the majority of circRNAs (9,129 of 9,799) contained repeat elements in their flanking introns (Figures 4D and S22). Upon further analysis we found that the repeat families DNA/hAT, LTR, and nMITE were the most frequently and significantly occurring inverted-repeat element pairs in circRNA flanking introns (Figure 4D, see STAR Methods).

To discern differentially expressed circRNAs, we grouped the total RNA-seq samples into five groups: a mix of adult tissues (see STAR Methods), adult forelimb, adult limb stump, adult regenerating limb, and larvae. We performed principal-component analysis (PCA) of expressed circRNAs, which revealed

**Figure 3. Genome expansion, repeat element composition, and regeneration-associated expression of hATs and domesticated hATs in *P. waltl***

(A) Bar plots showing the contribution of the main repeat element classes to each of the indicated genomes. Note the abundance of DNA repeat elements in *P. waltl*.

(B) Heatmap representing percentage contribution of repeat element families to the genome of each indicated species. Families are grouped by repeat element type.

(C) Relative contributions of hAT families to the total repeat element repertoire of the *P. waltl* genome, expressed as a percentage of all repeat element based on sequence length.

(D) Expression of indicated hAT elements in PacBio Iso-seq-derived transcriptomes from *P. waltl* limb blastema, brain, and spleen.

(E) Differential expression of hATs (left) as well as domesticated hATs, *Zbed1*, *Zbed4*, *Zbed6*, and *Zbed8* (right), during limb regeneration in *P. waltl*, based on normalized and centered RNA-seq counts for the indicated conditions. Only genes whose differential expression is significantly altered (between 0 and 3 dpa or 0 and 7 dpa) are depicted. Color key represents *Z*-score values.

**A**

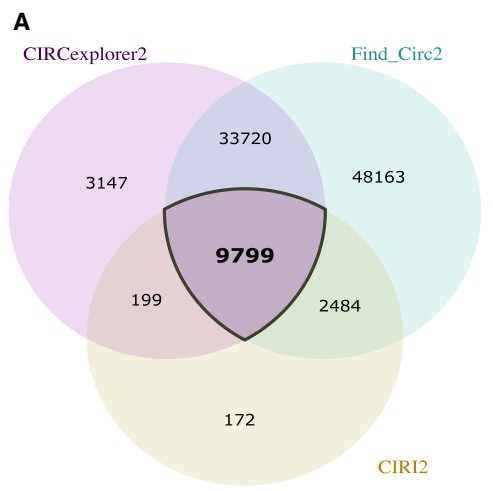

**B**

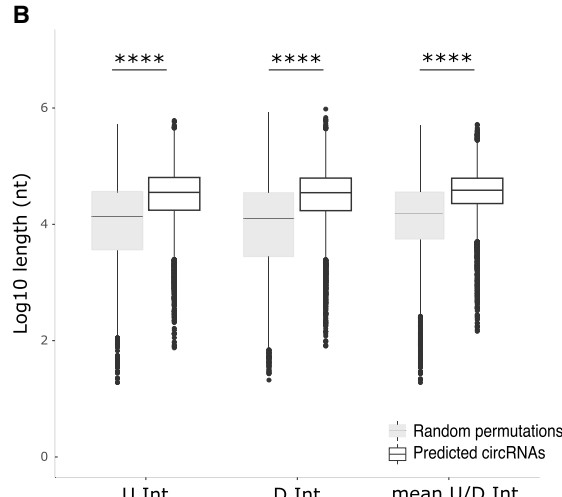

**C**

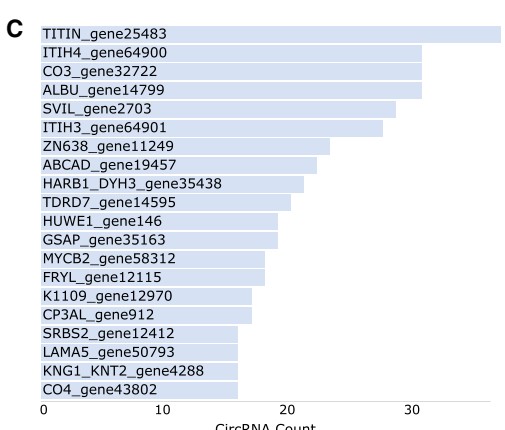

**D**

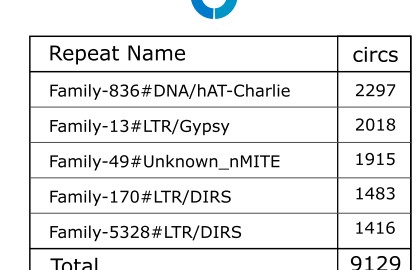

| Repeat Name | circs |
|---|---|
| Family-836#DNA/hAT-Charlie | 2297 |
| Family-13#LTR/Gypsy | 2018 |
| Family-49#Unknown_nMITE | 1915 |
| Family-170#LTR/DIRS | 1483 |
| Family-5328#LTR/DIRS | 1416 |
| Total | 9129 |

**E**

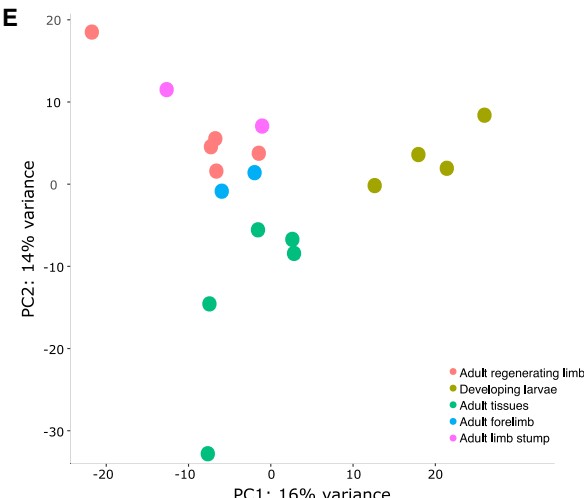

**F** Developing larvae vs Adult regenerating limb

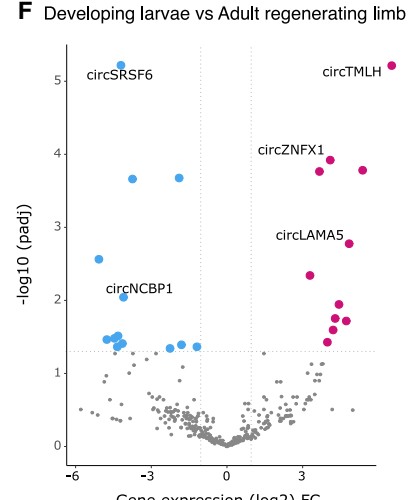

(legend on next page)

distinct expression profiles, indicating tissue-specific regulation (Figures 4E and S23). Given the unique regenerative capabilities of newts, we compared circRNA expression in the regenerating limb versus several other tissues. The data showed tissue- and context-specific expression profiles, in line with the notion that limb regeneration is driven by distinct molecular mechanisms compared to both development and homeostasis[45,46] (Figures 4F and S24).

These data collectively provide evidence for intronic repeat elements driving newt-specific circRNA biogenesis and indicate regeneration-specific regulation of their expression.

### miRNAs are embedded in repeat elements

Mature miRNAs are short non-coding RNAs (~22 nt) cleaved from a stem-loop precursor transcript that can be captured by small RNA-seq techniques.[47,48] Since miRNAs have been shown to regulate stem cell fate in general and regeneration processes in salamanders in particular,[49,50] we set out to generate a *P. waltl* miRNA atlas through global identification of miRNAs (Figure 5A). To specifically enrich for miRNA reads among all small RNAs, we parsed 10 previously generated small RNA-seq datasets using the miRTrace software[51] prior to miRNA discovery by the miRDeep2 algorithm[52] (Figure 5A and Data S6). This yielded 855 high-confidence miRNA candidates throughout the genome, the majority of which (639, ~75%) were not reported in the previous annotation[11] (Data S7 and Table S9). The 855 miRNAs displayed diverse tissue-wide expression (Figure 5B). Analysis of genomic distribution indicated that miRNAs were predominately located in either intergenic (48%) or intronic regions (mirtrons) (41%), with a small number residing within exons (11%) (Figure 5C). In addition, we found over half of the identified miRNAs to be embedded in, or within 100 bp of, a repeat element (486 miRNAs, ~57%, Figure 5D). Of those embedded in repeat elements, miRNAs are hosted most frequently and significantly in class I LTR family repeats (214/486 miRNAs, 44%, $p < 0.019$, Table S9). Leveraging the present chromosome-level assembly, we further traced the locations of all miRNAs, highlighting a widespread distribution across all 12 chromosomes (Figure 5E). As an additional, independent approach to small RNA-seq-based miRDeep2 analysis, we ran MirMachine, which predicts miRNAs in the genome based on homology to evolutionarily conserved miRNA families.[53] Using 75 metazoan species, MirMachine reported ~500 miRNAs, of which 171 overlapped with miRDeep2 predictions (Figure 5A and Tables S10 and S11). In sum, the data represent the first in-depth annotation and chromosomal locali-

zation of miRNAs for a salamander species and reveal their relations to repeat elements.

### Expansion of embryonic stem cell-specific cell cycle-regulating miRNAs and species-specific miRNAs

Previous work[11] identified a large expansion of miR-427/430/302 containing the target-binding seed site AAGUGC, a signature of the embryonic stem cell-specific cell cycle-regulating miRNAs (ESCC-miRNAs).[54] To detect if expansion of miR-427 in *P. waltl* is in scale with genome size (~20.3 Gb), we compared predictions from MirMachine, which has been used to annotate miRNAs in several other large genomes.[53] MirMachine detected a higher number of *P. waltl* miR-427 paralogs (~300 copies, Table S11) than in species with even larger genomes, including *A. mexicanum* (~30 Gb) and *P. annectens* (~40 Gb) (Figure 6A). This indicates that increased genome size does not linearly correlate with increased copies of miR-427 and that this miRNA is uniquely expanded in *P. waltl.* We have focused on 43 ESCC-miRNA genes whose annotation is supported by mature miRNA expression (Table S12). This consists of 25 miRNA precursors with homology to *X. tropicalis* miR-427, 10 homologous to miR-93b, and 8 with miR-17 homology (Table S13). Strikingly, there are 14 distinct mature sequences among these 25 miR-427 precursors in *P. waltl*, compared to only 3 in *Xenopus* and 5 in zebrafish, despite reporting of ~80[55] and ~100[56] miR-427 precursors in these species, respectively (Table S13). Similarly, we observed an increased variation among mature sequences for the other ESCC-miRNAs, miR-93b and miR-17 (Table S13). To determine if the observed expansions of ESCC-miRNAs are representing true duplication events and not caused by genome assembly or annotation artifacts, we performed multiple sequence alignment for all 43 ESCC-miRNAs (Figures 6B and 6C). We found that, while the seed region is highly conserved across these miRNAs, there is divergence in the stem-loop precursor sequences that suggests genuine duplication events with strong evolutionary pressure to specifically preserve the seed sequence.[57]

We also found that ~84% (36 miRNAs) of ESCC-miRNAs resided in intergenic regions of the *P. waltl* genome and 12% (5 miRNAs) were mirtrons whose flanks were defined by splicing (Figure 5C and Table S12). These mirtrons locate to highly conserved genes involved in processes ranging from chromosome maintenance (e.g., *Mcm7*) to stemness (e.g., *Mycb2*). In the case of *Mcm7*, we observed intronic hosting of the ESCC-miRNA, miR-93a, to be conserved in *X. tropicalis.*[55] In addition,

---

**Figure 4. Characterization of the circular RNA transcriptome in *P. waltl***

(A) Venn diagram of intersection between different circRNA prediction programs. A total of 9,799 circRNA candidates were detected by all three programs.

(B) Longer introns flank predicted circRNAs compared to introns flanking randomly generated backspliced junctions. The mean length of up- and downstream flanking introns for predictions was 49,805 nt and for random permutations was 29,619 nt. U Int, upstream flanking introns; D Int, downstream flanking introns; ****$p < 0.0001$ (Welch's t test).

(C) Bar plot of "hotspot" genes, which host 15 or more circRNA isoforms.

(D) Schematic of circRNA formation promoted by inverted-repeat elements in flanking introns (top). Table of repeat elements that frequently overlap with down- and upstream introns flanking *P. waltl* circRNAs (bottom).

(E) Principal-component analysis plot based on circRNA expression across tissue groups. Tissue groups include adult tissue (brain, eyes, heart, liver, and lung), adult forelimb ($n = 2$), adult limb stump (D0 limb stump tissue), developing larvae (larvae at limb bud stage, late embryo stages 22 + 25), and adult regenerating limb ($n = 2$ of 3 dpa and $n = 3$ of 7 dpa). dpa, days post-amputation.

(F) Volcano plot of differentially expressed circRNAs in adult regenerating limb (3 and 7 dpa) compared to developing larvae (limb bud stage and s22 + s25). dpa, days post-amputation.

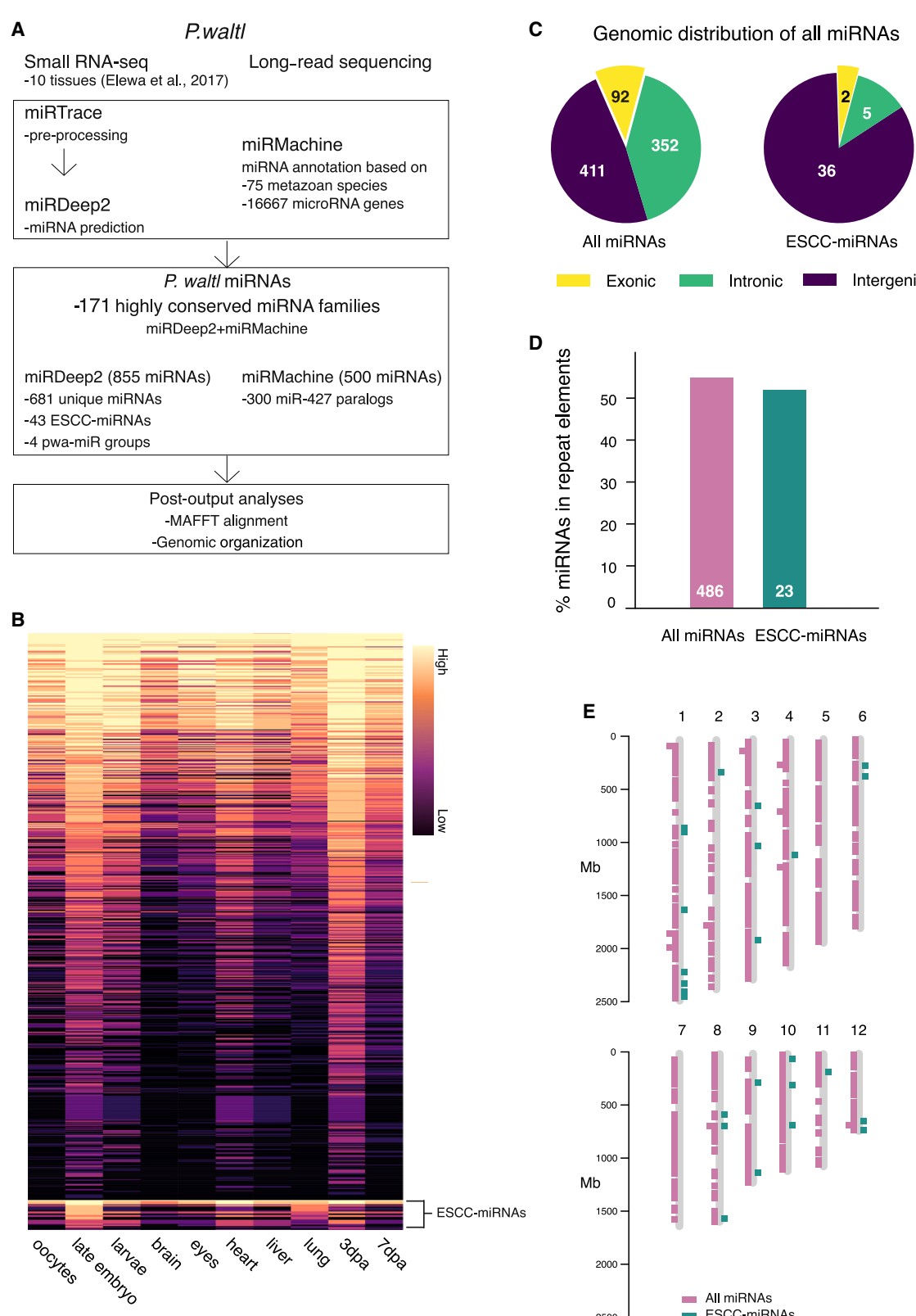

(legend on next page)

chromosome mapping of ESCC-miRNAs identified a clustering of miR-427 localized on chromosome 10 (Figure 5E and Table S12). Similar clustering has been described on chromosome 4 in both zebrafish and human and on chromosome 3 in *X. tropicalis*.[58] Another one-third of *P. waltl* miR-427 copies are widely distributed throughout the genome (Figure 5E and Table S12). We hypothesized that such an expansion of ESCC-miRNAs in the *P. waltl* genome may be due to embedding of miRNAs in repeat elements that may facilitate its duplication. Indeed, over half of the miRDeep2-predicted ESCC-miRNAs and almost all ~300 putative copies of miR-427 identified by MirMachine are found in repeat elements, most commonly within class I LTR/Gypsy elements (Figure 5D and Tables S11 and S12). Importantly, mapped small RNAs provide strong evidence that miRNAs in repeat elements are actively expressed; e.g., miR-427 (within ClassI_LTR_Gypsy) and miR-93 (within LTR/Gypsy) were detected in sequencing 6,043 and 330 times, respectively (Table S12).

As the expansion and expression of specific miRNAs could indicate a biological function,[48] we searched for additional mature miRNA sequences with multiple genomic copies in *P. waltl*. This search identified four groups of duplicated miRNA precursors, each with its own identical mature sequence (Figure S25 and Table S14). None of the four duplicated mature sequences has close homology with any miRNAs in either miRBase or mirGeneDB, implying these may be species specific; hence, we have denoted them as pwa-miRs. To ascertain if these novel mature sequences are *bona fide* miRNAs, we determined whether they are present in multiple small RNA-seq libraries (see STAR Methods and Table S15). The analysis provided evidence that multiple copies in each group are actively expressed and processed into mature miRNAs, supported by reads uniquely mapping to different genomic loci. We observed these reads to be present in several organs/tissue samples (Figure S26). Importantly, we detected mature pwa-miR sequence reads from several loci in the regenerating 3 and 7 dpa limb blastemas (Table S15 and Figure S26). Expansion of pwa-miRs through recent transposition events is supported by significant localization of numerous pwa-miR genomic copies locating to LTR (70%) repeat elements ($p < 2.8e-15$, Table S14). Altogether, we report a comprehensive characterization of both the ESCC-miRNAs and the pwa-miRs, highlighting conserved as well as unique genomic expansion profiles.

## DISCUSSION

How repeat elements shape genomes and, in particular, noncoding genomic regions has been difficult to address, especially in very large genomes. We show here that repeat elements make up 74% of the genomic content in *P. waltl*. Interestingly, the contribution from DNA transposable elements is notably larger than in all other salamander genomes studied to date.[19,34,59,60] In addition, we find that many DNA hAT elements are actively transcribed in various tissues, and members of this transposon superfamily as well as hAT-derived genes appear to be regulated during limb regeneration in adult newts. ZBED, hAT-derived genes, are of particular interest given that *Zbed4* is upregulated in lungfish tail blastema.[61] The class I repeat element LINE-1 is shown to be activated during axolotl limb regeneration.[62] Hence, hATs, members of the class II repeat elements, may represent previously unrecognized molecular components of vertebrate limb regeneration. However, it is known that repeat elements can be transcribed as a by-product of open chromatin and active expression of nearby transcripts.[63,64] Therefore, specific mechanisms that may regulate repeat element expression and their potential role in *P. waltl* regeneration require further investigation.

Intronic regions with inverted repeats derived from recently active transposable elements have been shown to promote the formation of circRNAs in a species-specific manner.[13] This highlights the importance of investigating the unique circRNA transcriptome of the newt, given the vast distribution of transposable elements throughout the genome that have contributed to intronic expansion. We utilized the assembled genome to detect and characterize *P. waltl* circRNAs. Even in total RNA-seq libraries, which are not enriched for circRNAs, we detected over 9,000 circRNAs. We therefore expect that this represents only the most abundant circRNAs,[38] while the real number of circRNAs in *P. waltl* is likely higher. We further identified inverted-repeat elements in intron pairs flanking circRNA back-spliced junctions, most frequently hATs, LTRs, and nMITEs. These data underscore yet another role of hAT elements as potentially the greatest driver of circRNA biogenesis in this species. Our findings also corroborate the hypothesis of repeat-element involvement in circRNA formation and indicate specific expression profiles in the regenerating salamander limb. Why hAT elements contribute significantly to the formation of circRNAs as opposed to the most abundant DNA Harbingers, among the class II repeat elements, remains unknown. Nevertheless, our observations underscore the importance of further exploring the putative functions of circRNAs in salamanders, especially in the context of limb regeneration.

The complete chromosome-level annotation of miRNAs in *P. waltl* is the first reported among giant genomes. A high abundance of repeat elements may lead to artifacts in genome assemblies and false positive prediction of inverted repeats, such as stem-loop miRNA structures, creating the potential for an inflated number of annotated miRNAs. Our approach of utilizing MirMachine allowed us to identify 171 conserved miRNA families

---

**Figure 5. Annotation and genomic organization of miRNAs in *P. waltl***

(A) Pipeline for annotation of miRNAs.

(B) Heatmap of normalized mature miRNA read counts across tissues for all miRDeep2 candidates. Forty-three ESCC-miRNAs are highlighted.

(C) Pie chart of the genomic localization for all 855 miRNAs and the 43 ESCC-miRNAs.

(D) Bar plot representing the percentage of miRNAs that are embedded in or located within 100 bp of a repeat element.

(E) Distribution of all the *P. waltl* miRNAs at the chromosome level. All 855 miRNA genomic loci are indicated on the left and the 43 ESCC-miRNAs are indicated on the right.

ESCC, embryonic stem cell-specific cell-cycle regulating.

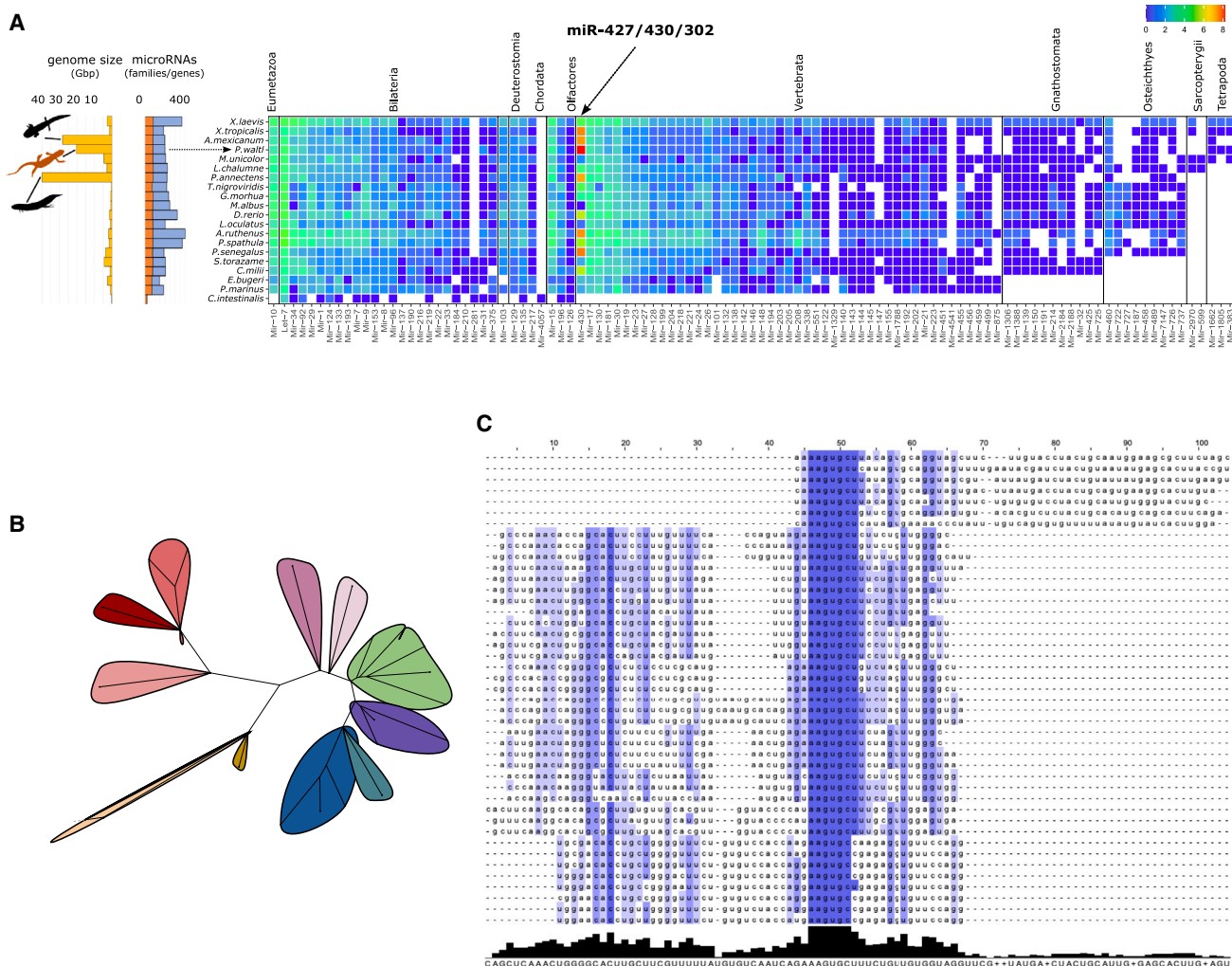

**Figure 6. Conserved miRNA paralogs and multiple alignment analyses of *P. waltl* ESCC-miRNAs**

(A) Heatmap of conserved miRNA paralogs across species of various genome sizes. Color scale corresponds to the number of paralogs (log2) in each organism, and arrows indicate miR-427/430/302 and *P. waltl*.

(B) Unrooted phylogenetic tree of the ESCC-miRNA precursors. Each spoke represents one ESCC-miRNA precursor, where greater distance between spokes indicates greater sequence diversity.

(C) MAFFT alignment of all 43 ESCC-miRNAs containing the conserved AAGUGC seed site on either the 5′ or the 3′ arm. Darker blue indicates higher conservation of bases. MAFFT, multiple alignment using fast Fourier transform.

in *P. waltl* based on 75 metazoan species.[53] These 171 miRs were further supported by miRDeep2[52] predictions that account for mature miRNA expression. We also report an additional 681 putative miRNAs predicted by miRDeep2. While some miRNA families are highly conserved across species, making them ideal phylogenetic markers,[65] species-specific variations have been reported among conserved miRNA families. One such example is the genomic expansion of miR-427 precursor sequences. Although zebrafish has ~100 precursor copies of miR-430 (corresponds to miR-427 in amphibians),[56] and ~80 precursor copies of miR-427 are found in *Xenopus*,[55] the greater diversity among ESCC-miRNA mature sequences in *P. waltl* indicates a more ancient expansion. More divergence among mature sequences could also allow for substantially increased diversity in ESCC-miRNA regulatory targets and novel functionality in

the newt.[66] Both ESCC-miRNAs and the four groups of pwamiR copies were predominately found embedded in LTR repeat elements, class I, providing a plausible mechanism behind the miRNA expansion in *P. waltl*. Interestingly, class I was also reported as a major source of transposable-element-derived miRs in humans.[14,67] LTR repeat elements are not the only major contributor to the *P. waltl* genome yet contained a significant number of miRNAs. Although the molecular underpinning of this specific distribution pattern is not known at present, our findings support the hypothesis that novel miRNAs can originate from inverted repeats with the potential to form stem-loop structures that evolve into functional miRNA genes.[21,68] The results confirmed that at least a subset of miRNAs in repeat elements is actively expressed, and their detection in the regenerating limb suggests that species-specific miRNAs may contribute to

unique regeneration mechanisms in *P. waltl.* The identical mature pwa-miR sequences and the highly similar precursor sequences indicate a relatively recent expansion.[57] Unlike ESCC-miRNAs, conservation of the seed sequence and divergence of the terminal loop likely have yet to take place in pwa-miRs. Nevertheless, their genomic expansion and expression during limb regeneration might indicate a role for pwa-miRs in newts' regenerative capacity.

Vertebrate species are characterized by a large diversity in regenerative capabilities, including key differences in regeneration mechanisms between the relatively closely related salamander species, the newt and the axolotl.[45,69] Comparative analyses of the genomes of the two species show notable variations, including repeat element class superfamily content, MHC gene organization, and overall miRNA landscape. The findings regarding the expansion of the ESCC-miRNAs and the existence of pwa-miRNAs provide new avenues to investigate the molecular basis of differences, for example, in how regenerative progenitor cells are generated in these two salamander species.

From a broader perspective, the data uncover how repeat elements shape a gigantic genome, serving as common denominators between genome expansion, facilitation of circRNA formation, and evolution of miRNAs from stem-loop structures. Furthermore, the chromosome-scale assembly of *P. waltl* provides a useful resource that facilitates our understanding of vertebrate chromosome evolution, not the least, variations among syntenic blocks and gene content. Our results thus allow for the exploration of species-specific innovations and evolutionarily shared molecular responses to major injuries, knowledge necessary to identify mechanisms that promote or counteract successful regeneration.

## Limitations of the study

The contiguity of the *P. waltl* genome we present here is the most complete among all giant genomes sequenced to date. Achieving this was challenging, because repeat elements constitute the vast majority of the genome. We could accurately classify several repeat element families and confirm that the increased genome size is linked to repeat elements and not duplication events. We acknowledge, however, that the repeat element library we provide here is not manually curated, which ultimately may affect the exact estimates of repeat element distributions. Due to the sheer number of repeat elements, a manual classification is outside the scope of this paper. Nevertheless, the data presented here represent a key resource that can be further utilized and built upon by the scientific community. We also provide evidence that repeat elements are active during newt limb regeneration and play a key role in the formation of non-coding RNAs such as circRNAs and miRNAs. Functional studies will be required to discern the exact mechanisms and the implications of this expression pattern during regeneration.

## RESOURCE AVAILABILITY

### Lead contact

Requests for further information and resources should be directed to and will be fulfilled by the lead contact, András Simon (andras.simon@ki.se).

### Materials availability

This study did not generate new unique reagents.

### Data and code availability

Genome and annotation files are available through the Max Planck Digital Library (https://doi.org/10.17617/3.90C1ND) and through NCBI under BioProject PRJNA847026. Iso-seq data for spleen and limb blastema are available at NCBI, BioProject PRJNA1193474 and PRJDB19556, respectively. All original code has been deposited at Zenodo at https://doi.org/10.5281/zenodo.14505834.

## ACKNOWLEDGMENTS

We thank Elly Tanaka for valuable discussion at the initiation of this project and Olga Vinnere Pettersson and Silke Winkler for advice on sequencing strategies. Storage and handling of sequencing data were enabled by resources provided by the Swedish National Infrastructure for Computing (SNIC) at UPPMAX, partially funded by the Swedish Research Council through grant agreement 2018-05973; the DRESDEN Concept Genome Center, part of the technology platform of the CMCB at TU Dresden, supported by DFG (INST 269/768-1); the MPI-CBG computing cloud; and the Center for Information Services and High-Performance Computing (ZIH) at Technische Universität Dresden. Work performed at NGI/Uppsala Genome Center has been funded by RFI/VR and Science for Life Laboratory, Sweden. We thank Miho Kiyooka and Wei Chen for blastema Iso-seq library preparation and Sequel sequencing in the National Institute for Genetics (Japan) and Dominick Kruger, Beate Gruhl, and Anja Wagner (CRTD, Germany) for newt husbandry at TU in Dresden and Chad Donaldson at KI in Stockholm. The work was funded by the Swedish Research Council (2023-02820), Cancerfonden (22 2424 Pj), European Research Council (951477), Knut and Allice Wallenberg Foundation (project grant 2018-2022), and Karolinska Institute to A.S.; DFG (22137416, 450807335, and 497658823) grants and TUD and CRTD funds to M.H.Y.; Swedish Research Council (2020-01486), Cancerfonden (23 3047 Pj), and Knut and Alice Wallenberg Foundation to N.D.L.; and JSPS KAKENHI grant number JP16H06279 (PAGS) to K.-i.T.S. and A.T.

## AUTHOR CONTRIBUTIONS

Conceptualization: the *Pleurodeles waltl* genome project was initiated by A.S.; assembly, A.S., M.H.Y., and N.D.L.; annotation, A.S., M.H.Y., and N.D.L.; synteny, M.H.Y. and N.D.L.; transposable element analyses, M.H.Y.; circRNA analyses, K.M.; miRNA analyses, A.S. and K.M. Methodology: assembly, T.B. and E.O.; annotation, T.B., A.E., and A.P.; synteny analyses, T.B. and L.R.; transposable element analyses, S.I. with input from T.B., A.E., and N.D.L.; circRNA analyses, K.M.; miRNA analyses, K.M., M.R.F., and B.F. Formal analysis: assembly and annotation, T.B., A.E., A.P., and E.O.; synteny analyses, T.B., L.R., and N.D.L.; transposable element analyses, S.I. with input from T.B., A.E., and N.D.L.; circRNA analyses, K.M.; miRNA analyses, K.M., M.R.F., and B.F. Investigation: assembly and annotation, sample preparation by A.J.A., E.S., and N.D.L.; synteny analyses, T.B., L.R., C.R.O., and N.D.L.; transposable element analyses, S.I.; circRNA analyses, K.M.; miRNA analyses, K.M., M.R.F., and B.F. Visualization: assembly, T.B.; synteny analyses, T.B., L.R., and N.D.L. with input from K.M.; transposable element analyses, S.I.; with input from T.B. and A.E.; circRNA analyses, K.M.; miRNA analyses, K.M., M.R.F., and B.F.; graphical abstract, A.J.A. and K.M. Resources: assembly, A.S. and M.H.Y.; annotation, A.S., M.H.Y., M.S., K.-i.T.S., T.H., and A.T.; synteny, M.H.Y. and N.D.L.; transposable element analyses, M.H.Y.; circRNA analyses, A.S.; miRNA analyses, A.S. Writing – original draft: K.M., A.S., and M.H.Y.; writing – review & editing, K.M., A.S., M.H.Y., N.D.L., and all authors. Supervision: assembly, M.H.Y. and N.D.L.; annotation, A.S., M.H.Y., and N.D.L.; synteny, M.H.Y. and N.D.L.; transposable element analyses, M.H.Y. with input from A.E. and N.D.L.; miRNA analyses, A.S. and M.R.F. Project administration: A.S., M.H.Y., and N.D.L. Funding acquisition: A.S., M.H.Y., and N.D.L.

## DECLARATION OF INTERESTS

M.H.Y. is a co-founder of Faunsome, Inc.

## CellPress

## Cell Genomics

## STAR★METHODS

Detailed methods are provided in the online version of this paper and include the following:

- ● KEY RESOURCES TABLE
- ● EXPERIMENTAL MODEL AND SUBJECT DETAILS
  - ○ Animal procedures
- ● METHOD DETAILS
  - ○ Genome sequencing
  - ○ Hi-C
  - ○ Iso-seq
  - ○ Genome assembly and scaffolding
  - ○ Genome annotation
  - ○ Intron counting
  - ○ Repeat element annotation and analysis
  - ○ *P. waltl* transcriptome analysis
  - ○ Reverse transcription quantitative real-time PCR
  - ○ circRNA characterisation
  - ○ miRNA identification and genomic distribution
- ● QUANTIFICATION AND STATISTICAL ANALYSIS

## SUPPLEMENTAL INFORMATION

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

# STAR★METHODS

## KEY RESOURCES TABLE

| REAGENT or RESOURCE | SOURCE | IDENTIFIER |
|---|---|---|
| **Critical commercial assays** | | |
| Monarch® HMW DNA Extraction Kit | New England Biolabs | Cat#T3010; Cat#T3060 |
| SMRTbell Express Template Prep Kit 2.0 | Pacific Biosciences | Cat#PN 101-853-100 Version 05 |
| ARIMA-HiC+ High Coverage Kit | Arima Genomics | Cat#A101030-ARI |
| Total RNA purification kit | Norgen Biotek | Cat#17200 |
| **Deposited data** | | |
| PacBio HiFi | This Paper | SRA: SRR22542941 |
| Hi-C | This Paper | SRA: SRR22542940 |
| *Pleurodeles waltl* genome assemblies | This Paper | GenBank: GCA_026652325.1, GCA_031143425.1, GCA_031142525.1 |
| *Pleurodeles waltl* genome assemblies and annotations | This Paper | https://doi.org/10.17617/3.90C1ND |
| *Pleurodeles waltl* transcriptomes | Elewa et al.[11] & Matsunami et al.[29] | BioProjects: PRJNA353981, PRJDB7442 |
| *Pleurodeles waltl* iso-seq | This Paper | BioProjects: PRJNA1193474, PRJNA858999, PRJDB19556 |
| *Protopterus annectens* genome | Wang et al.[24] | GenBank: GCA_019279795.1 |
| *Ambystoma mexicanum* genome | Schloissnig et al.[26] | GenBank: GCA_002915635.3 |
| *Lepisosteus oculatus* genome | Braasch et al.[31] | GenBank: GCA_000242695.1 |
| **Software and algorithms** | | |
| ccs v6.0.0 | PacBio | https://github.com/PacificBiosciences/ccs |
| Hifiasm v0.16.0 & v0.16.1 | Cheng et al.[22] | https://github.com/chhylp123/hifiasm |
| purge-dups v1.2.3 | Guan et al.[23] | https://github.com/dfguan/purge_dups |
| Salsa2 v2.2 | Ghurye et al.[27] | https://github.com/marbl/SALSA |
| Pbmm2 v1.3.0 | PacBio | https://github.com/PacificBiosciences/pbmm2 |
| bwa-mem v0.7.17 | Li - 1 | https://github.com/lh3/bwa |
| minimap2 v2.24 | Li - 2 | https://github.com/lh3/minimap2 |
| Picard MarkDuplicates 2.22.6 | https://broadinstitute.github.io/picard/ | https://github.com/broadinstitute/picard/tree/master |
| table2asn | NCBI | https://ftp.ncbi.nlm.nih.gov/asn1-converters/by_program/table2asn/ |
| blastp v2.11.0 | NCBI | https://ftp.ncbi.nlm.nih.gov/blast/executables/blast+/ |
| hmmscan v3.3.2 | http://hmmer.org/ | http://hmmer.org/ |
| transdecoder v5.5.0 | https://github.com/TransDecoder/TransDecoder | https://github.com/TransDecoder/TransDecoder |
| RepeatModeler v2.0.2 | http://www.repeatmasker.org | https://github.com/Dfam-consortium/RepeatModeler/tree/master |
| RepeatMasker v4.1.2 | http://www.repeatmasker.org | https://github.com/Dfam-consortium/RepeatMasker |
| agat v1.0.0 | Dainat et al.[70] | https://github.com/NBISweden/AGAT |
| Trinotate v3.2.2 & 4.0.1 | Bryant et al.[71] | https://github.com/Trinotate/Trinotate |

*(Continued on next page)*

*ll* CellPress

| *Continued* | | |
|---|---|---|
| REAGENT or RESOURCE | SOURCE | IDENTIFIER |
| Ucsc-tools | UCSC | https://hgdownload.soe.ucsc.edu/admin/exe/ |
| BUSCO v5.7.1 | Manni et al.[72] | https://gitlab.com/ezlab/busco |
| odp v0.3.3 | Schultz et al.[73] | https://github.com/conchoecia/odp |
| OMArk 2.4.0 | Nevers et al.[74] | https://github.com/DessimozLab/OMArk |
| OMAmer v0.3.0 | Rossier et al.[75] | https://github.com/DessimozLab/omamer |
| Pandas v2.2.2 | The pandas development team | https://doi.org/10.5281/zenodo.3509134 |

## EXPERIMENTAL MODEL AND SUBJECT DETAILS

### Animal procedures

All animal experiments were performed in accordance with the guidelines of local, European and Japanese ethical permits. Adult Iberian Ribbed newts (*P. waltl*) were obtained from the aquatic animal facility in Karolinska Institutet (genome sequencing) and CRTD-Center for Regenerative Therapies Dresden (RT-qPCR analysis), where they were bred, raised and kept in captivity for generations as described.[76] Animal procedures at CRTD were conducted in compliance with the German Animal Welfare Act and legislation from the state of Saxony, under the licence TVV 36/2018. For genome sequencing, a female newt was used, as in this species the females represent the sex that display a heterogametic ZW genotype involved in sex determination.[77] To ensure that sequencing would be performed in a diploid animal, we analysed tail tip tissue from a tail clip from six adult females. The tissue collected was fixed, processed, sectioned at 30μm, and stained with DAPI to highlight the DNA according to.[7] The 2n DNA content was confirmed by analysing both the expected size of the nuclei as well as chromosome quantification in cartilage cells found in the M-phase of the cell cycle (2n = 24 chromosomes,[11] (Figure S1). A 14.1cm-long adult female weighing 17.93g was selected for tissue collection. Tissues from the same animal were used for genome sequencing, Hi-C as well as brain and spleen Iso-Seq. After the tail clip was allowed to regenerate, the animal was deeply anaesthetised in 0.1% tricaine (MS-222, Sigma, pH=7) and sacrificed by decapitation. Immediately after, tissue dissection and collection were performed by three researchers working simultaneously to minimise sample degradation. The samples collected consisted of blood, brain, spinal cord, limb skeletal muscle, spleen, heart, and liver. Samples were snap-frozen in liquid nitrogen and stored at -80°C for further processing.

For blastema Iso-Seq preparation, adult newts were obtained from the Amphibian research centre at Hiroshima University, Japan, and their rearing and treatments were performed and approved in accordance with the Guidelines for the Use and Care of Experimental Animals and the Institutional Animal Care and Use Committee of National Institute for Basic Biology, Japan.

For RT-qPCR analysis of the indicated limb regeneration stages, newts were anesthetised in 0.03% benzocaine (Sigma) prior to limb amputation at the mid-humerus level. Animals were allowed to recover and regenerate at 20 °C, and limbs or blastemas collected and processed for RNA extraction as specified below. Experiments were performed in accordance with the laws and regulations of the State of Saxony.

## METHOD DETAILS

### Genome sequencing

The blood and spleen from a female, adult *P. waltl* was collected. Genomic DNA was extracted using the Monarch® HMW DNA Extraction Kit for blood and tissue (NEB #T3010 and NEB #T3060, respectively) as per manufacturer's protocol. DNA was then immediately frozen and stored at -20°C until further downstream processing. Input QC of the DNA was performed using Dropsense, Qubit and Femto pulse to evaluate concentration, purity and size. Sample libraries were prepared according to Pacbio's Procedure & Checklist – Preparing HiFi SMRTbell® Libraries using the SMRTbell Express Template Prep Kit 2.0, PN 101-853-100 Version 05 (August 2021) using the SMRTbell Express Template Prep Kit 2.0. Samples were sheared on Megaruptor 3 with speed setting 31 followed by 32. An Ampure bead purification was performed after the shearing. The samples were size selected using SageElf, according to Pacbio's protocol. Fractions 1-4 were used for sequencing. Quality control of sheared DNA and SMRTbell libraries was performed on Fragment analyzer, using the Large Fragment standard sensitivity 492 kit. Primer annealing and polymerase binding was performed using the Sequel II binding kit 2.2. They were sequenced on the Sequel II and IIe instrument, using the Sequel II sequencing plate 2.0 and the Sequel® II SMRT® Cell 8M, movie time 30 hours and pre-extension time 2 hours.

### Hi-C

Chromatin conformation capturing was performed using the ARIMA-HiC+ High Coverage Kit (Article Nr. A101030-ARI) following the user guide for animal tissues (ARIMA Document, Part Number: A160162 v00). In brief, 40 mg flash-frozen powdered heart tissue was chemically crosslinked. The crosslinked genomic DNA was digested with a restriction enzyme cocktail consisting of four restriction enzymes. The 5'-overhangs were filled in and labelled with biotin. Spatially proximal digested DNA ends were ligated and purified.

**Article**

The proximally-ligated DNA was then sheared and enriched for the ligated biotin containing fragments. Due to the expected genome size, two barcoded Illumina sequencing libraries were prepared following the ARIMA user guide for Library preparation using the Kapa Hyper Prep kit (ARIMA Document Part Number A160139 v00). The barcoded Hi-C libraries were sequenced on a NovaSeq6000 (NovaSeq Control Software 1.7.5/RTA v3.4.4) with a 151nt (Read1)-10nt (Index1)-10nt (Index2)-151nt (Read2) setup using 'NovaSeqXp' workflow in 'S4' mode flowcell. The Bcl to FastQ conversion was performed using bcl2fastq_v2.20.0.422 from the CASAVA software suite. The quality scale used was Sanger / phred33 / Illumina 1.8+.

### Iso-seq

Spleen Iso-Seq: Tissue was harvested from the spleen of the female adult *P. waltl* that was used for genome sequencing. Samples were snap-frozen in liquid nitrogen and stored at -80°C for further processing. Approximately 10 mg of each indicated tissue were pulverised in liquid nitrogen and used for RNA extraction by means of Total RNA Purification Kit (Cat. 17200 from Norgen Biotek) as per manufacturer's instructions. The pulverised spleen powders are mixed well with the β-mercaptoethanol containing buffer RL and the homogenates are further passing through the needle attached syringe for 5-10 times. The genomic DNA was removed via on column DNA removal using Norgen RNAse free DNAase kit (Cat. 25710). Input QC of the RNA was performed on the Agilent Bioanalyzer instrument, using the Eukaryote Total RNA Nano kit to evaluate RIN and concentration. RIN value obtained was above 9 for both samples. The sample libraries were prepared according to Pacbio's Procedure & Checklist – Iso-Seq™ Express Template Preparation for Sequel® and Sequel II Systems, PN 101-763-800 Version 02 (October 2019) using the NEBNext® Single Cell/Low Input cDNA Synthesis & Amplification Module, the Iso-Seq Express Oligo Kit, ProNex beads and the SMRTbell Express Template Prep Kit 2.0. 300 ng RNA was used as input material. The samples were amplified 12 cycles. In the purification of amplified cDNA the standard workflow was applied (sample is composed primarily of transcripts centred around 2 kb). Quality control of the SMRTbell libraries was performed with the Qubit dsDNA HS kit and the Agilent Bioanalyzer High Sensitivity kit. Primer annealing and polymerase binding was performed using the Sequel II binding kit 2.0. Libraries were sequenced on the Sequel IIe instrument, using the Sequel II sequencing plate 2.0 and the Sequel® II SMRT® Cell 8M, movie time=24 hours and pre-extension time=2 hours (1 SMRT cell per sample). Raw sequencing data was fed into IsoSeq3 (https://github.com/PacificBiosciences/IsoSeq), as per standard instructions, to generate fasta outputs of high-quality and low-quality reads. The high quality fasta output was taken for further use as part of the reference transcriptome.

Brain Iso-Seq: the dataset is published[28] and publicly available at NCBI (SRX16252717). Of note, the brain tissues correspond to the same individual described above.

Blastema Iso-Seq: Adult newts (1-year-old) were anaesthetised in 0.1% MS-222, and the forelimbs were amputated with a surgical knife at the middle of the forearm level. Blastema was collected at the late-bud blastema stage[78] under anaesthesia, and 4-6 blastemas were pooled for each library. Total RNAs were purified using RNeasy mini kit (Qiagen) or NucleoSpin RNA Plus XS (Takara Bio) kit. Sample libraries were prepared according to the following PacBio Iso-Seq protocols: Iso-Seq™ Template Preparation for Sequel Systems and Iso-Seq™ Express Template Preparation for Sequel and Sequel II Systems. One of four blastema libraries was treated with gRNA/Cas9 ribonucleocomplex which targets 20 transcripts highly expressed in blastema to deplete abundant transcripts *in vitro* after the first bead purification step, another was treated with tyr Cas9 RNP to act as negative control, and the other two samples were untreated (PRJDB19556). Libraries were sequenced on the Sequel I or II in National Institute for Genetics, Japan.

### Genome assembly and scaffolding

CCS reads (rq > 0.99) were called from the subreads.bam file using PacBio ccs (v6.0.0). Two contig assemblies were then created using hifiasm (v 0.16.0-r369) with arguments -l2 using the ccs reads as input. Purge-dups (v1.2.3) was then run on the primary contigs to create the set of contigs used for scaffolding. A further run of purge-dups on the combined assembly made up of the purged output from the primary contigs and the alt contigs from hifiasm created the alternate assembly.

To polish the assembled contigs from the primary assembly, all CCS reads were mapped to the contig assemblies using pbmm2 with arguments: –preset CCS -N 1 and variants were called using DeepVariant (v 1.2.0). Sites were filtered for those with 'genotype 1/1' to specify that all or nearly all reads support an alternative sequence at this position and a 'PASS' filter value to specify that the site passed DeepVariant's internal filters. Base errors were then corrected using bcftools consensus (v 1.12). Based on the QV values produced by merqury (v 1.0) this resulted in an assembly with QV 72.9 and QV 54.8 using PacBio CCS reads and Arima V2 Hi-C reads as a k-mer database, respectively. The discrepancy between these values can be attributed to the fact the CCS reads were used to assemble the genome itself, therefore overestimate the accuracy of the genome, whereas the Hi-C reads are not uniform and contain coverage gaps, resulting in an under-estimate. The "true" value is likely to lie between these two values, but without an independent sequencing dataset we are unable to estimate the true value.

For scaffolding, SALSA2 and the VGP's Arima mapping pipeline were run (https://github.com/VGP/vgp-assembly/blob/master/pipeline/salsa/arima_mapping_pipeline.sh). Briefly, Hi-C reads were mapped to the contigs using BWA-MEM (v 0.7.17-r1198-dirty).

Alignments were then filtered using the Arima filter_five_end.pl script (https://github.com/VGP/vgp-assembly/blob/master/pipeline/salsa/filter_five_end.pl) and removed potential PCR duplicates with Picard's MarkDuplicates (v 2.22.6).

The resulting read-sorted bed file was used as input for SALSA2 (v 2.2). A number of manual curation rounds were then performed to correct scaffolding errors and to scaffold those contigs which were not automatically scaffolded into the 12 chromosomes. To this end, cooler (v 0.8.11) and HiGlass (v 2.1.11) were used to visually inspect the Hi-C maps and SeqKit (v 0.13.2) was used to re-arrange contigs and scaffolds into chromosome-level scaffolds.

**Cell Genomics**

To generate two haplotype-phased assemblies, we used hifiasm (v 0.16.1) with both PacBio CCS and HiC reads as input with arguments:

-l2 –h1 hic_R1.fastq.gz –h2 hic_R2.fastq.gz

and subsequently ran purge-dups (v1.2.3) separately on the two haplotypes as above. Each haplotype was then scaffolded and manually curated separately as above, with the exception that yahs (v1.1a) was used to scaffold instead of SALSA2.

BUSCO scores were obtained for the three P. waltl genome assemblies (primary, hap1 and hap2) using ran BUSCO v5.7.1[72] in —miniprot mode using the vertebrata_odb10 database.

### Repeat masking

In order to mask the *P. waltl* genome a *de novo* repeat library was first created using RepeatModeler (v 2.0.2) with argument -LTRStruct. The resulting Pleurodeles-specific repeat library was combined with the Dfam repeat library for all ancestors of Pleurodeles with famdb.py (v 0.4.2):

famdb.py -i Dfam.h5 families –ancestors -f fasta_name "pleurodeles waltl" > pleurodelesWaltl_repeatLibrary.fasta

The combined repeat library was then used to mask the genome using RepeatMasker (v 4.1.2):

RepeatMasker -gff -xsmall -e crossmatch -libdir RepeatMasker-4.1.2-p1/Libraries -lib aPleWal_combinedRepLibraries.fasta -rmblast_dir rmblast-2.11.0/bin -crossmatch_dir /sw/bin aPleWal_genome.fasta

The resolution of element boundaries was not conducted due to extensive curation requirements.

### Genome annotation

### Transcriptome and transcript mapping

cDNA from *de novo* transcriptomes[5,24] and high-quality Iso-seq reads from three libraries (brain, spleen and blastema) were aligned to the *P. waltl* genome using minimap2 (v 2.24-r1122) and the following arguments:

-l200G -axsplice:hq –secondary=no -uf -G5m -a

Gff files from the resulting sam file were created using the following commands:

samtools view -b in.sam | samtools sort -o in.sorted.bam

bedtools bamtobed -bed12 -i in.sorted.bam > in.bed

bedToGenePred in.bed in.genepred

genePredToGtf \"file\" in.genepred in.gtf

genometools-1.6.1/bin/gt gtf_to_gff3 -tidy in.gtf | genometools-1.6.1/bin/gt gff3 -tidy -sort > out.gff

### Augustus

The gff files produced from mapping transcripts to the genome (see Transcriptome and transcript mapping section) were used as evidence for augustus (v 3.4.0) with the following arguments:

augustus –extrinsicCfgFile=extrinsic.M.RM.E.W.cfg –hintsfile=hints.gff –species=chiloscyllium pleurodeles.fasta > augustus. predictions.gff

*Chiloscyllium* was chosen as a model as Braker was not able to produce a *de novo* model for *P. waltl* due to the genome size causing errors in the GeneMark step.

### EvidenceModeler

The gff files produced from the last two sections were given to EvidenceModeler (v 1.1.1) to filter alignments for correct annotations with weights:

OTHER_PREDICITON rna 2

ABINITIO_PREDICTION AUGUSTUS 1

A chunk size of 30Mb and an overlap size of 10Mb were used to avoid the larger introns being removed due to default sizes used by EvidenceModeler.

As EvidenceModeler aims to only produce one isoform per gene, afterwards we included any transcripts that were present in at least two of the transcript evidences (2 transcriptome assemblies and 3 Iso-seq libraries).

### Gene nomenclature

Combining "high-confidence" transcripts with the output from EvidenceModeler resulted in 164,283 predicted isoforms including 56,783 conserved protein coding isoforms (i.e., with a significant homology hit E-value cutoff 10ˆ-10 with a protein in UniprotKB'sSwiss-prot database). For the purpose of gene nomenclature and downstream analysis, multiple isoforms belonging to the same gene were grouped under the same name using a custom python script (aPleWal.mergeisoforms.py), which took isoforms with consecutive numbers (e.g. gene47, gene48, gene49, gene50 and gene51) and the same homology hit (e.g. XYNB_NEOPA) and gave them the same gene name (e.g. XYNB.1). In the event that multiple genes shared the same UNIPROT ID, each additional gene was given a greater integer suffix (e.g. XYNB.2, etc.). Transcript isoforms are identified with an additional integer suffix after the gene name (e.g. XYNB.1.1, XYNB.1.2, etc.). The outcome of this process created 142,667 gene models of which 35,167 were conserved protein coding genes. Finally ncbi's table2asn function (https://ftp.ncbi.nlm.nih.gov/asn1-converters/by_program/table2asn/) was used to filter spurious annotations for features such as internal stop-codons or errors in frame resulting in our gff3 file: aPleWal1.anno.20220803.gff3.

### *Protein-based synteny*

Ribbon plots and identification of spatially-localised groups of genes of one-to-one orthologs between the *P. waltl* primary, axolotl, and gar assemblies were generated using the Oxford Dot Plot (ODP) tool.[73] ODP was run including only the chromosomes from each assembly, or in the case of the gar, only the linkage groups, keeping the longest isoform per transcript using agat_sp_keep_longest_isoform.pl from agat v1.0.0[70] and only plotting those orthologs identified with a corrected p-value less than 0.01. ODP was run using the arguments:

    search_method: diamond
    duplicate_proteins: "pass"
    num_permutations: 1000000
    chr_sort_order: optimal-top
    plot_LGs: True
    plot_sp_sp: True
    plot_all: False

The identified best-reciprocal-hit orthologs between the Chordate Linkage Groups (CLGs)[30] in newt, axolotl and gar were plotted relative the locations of the identified orthologs in the respective genomes or linkage groups. Those groups of genes found to be co-localised with Fisher's Exact Test (FET) p-value less than 0.05 were highlighted as belonging to syntenic blocks, either plotting those orthologs in bold (Figure 2C) or being included in the lists of syntenic regions (Tables S2–S6). Orthologs between newt and axolotl and axolotl and gar and identified in one of the 17 CLGs were then visualised as ribbon plots (Figure 2D) showing the locations of orthologs in newt, axolotl and gar and their respective CLG.

Synteny plots between assemblies were created by aligning protein sequences from *P. waltl* annotation aPleWal1.anno.20220803.gff3 against protein sequences downloaded for *Homo sapiens* (grch38), axolotl -*Ambystoma mexicanum*- (AmexT_v47-AmexG_v6.0-DD), *Protopterus annectens* (GCF_019279795.1_PAN1.0), gar - *Lepisosteus oculatus*- (Lepisosteus_oculatus.LepOcu1) and *Xenopus tropicalis* (GCF_000004195.4_UCB_Xtro_10.0) using blastp (v 2.11.0+) and vice-versa only allowing max_target_seqs=1. In instances where the same transcripts were mapped against each other one-to-one, these locations in the two genomes were included in the synteny plots based on locations in the gff files. A detailed pipeline is available in Data S8.

### *Annotation v2*

A second version gene annotation was generated using purely transcript data:

RNA-seq data from BioProject PRJNA353981 were aligned to the genome using hisat2 (v2.2.1) cDNA and iso-seq were aligned to the genome using minimap2 as above and bam files were filtered using samtools view and the following arguments filtering for high-quality alignment (Q60) and removing any unpaired, secondary, or supplementary alignments:

    -F 3844 -q60

The merged bam file was used as input to stringtie (v2.2.1) (https://www.ncbi.nlm.nih.gov/pmc/articles/PMC4643835/) and coding sequences were identified using TransDecoder (v5.5.0) ( https://github.com/TransDecoder/TransDecoder) after first identifying protein homolog regions mapping to the Swissprot and Pfam databases using blastp (v2.11.0) (https://bmcbioinformatics.biomedcentral.com/articles/10.1186/1471-2105-10-421) and hmmscan (v3.3.2) (https://www.ncbi.nlm.nih.gov/pmc/articles/PMC3125773/) respectively.

Annotations with in-frame or early stop codons were removed using ncbi's table2asn (The NCBI C++ Toolkit (https://ncbi.github.io/cxx-toolkit/) by the National Center for Biotechnology Information, U.S. National Library of Medicine; Bethesda MD, 20894 USA) program and then orthologs to known proteins in Swissprot and Pfam were identified by mapping as above and filtering using Trinotate (v3.2.2) ( https://pubmed.ncbi.nlm.nih.gov/28099853/) resulting in annotation file: aPleWal.anno.v2.20220926.gff3. This annotation resulted in 65,597 gene models, 174,782 transcripts and 14,752 single exon genes.

Functional annotations were generated by creating a transcriptome using Trinotate_GTF_or_GFF3_annot_prep.pl from Trinotate (v4.0.1) and the aPleWal.anno.v2.20220926.gff3 and aPleWal1.pri.20220803.fasta files.

### *Microsynteny*

We generated a *P. waltl* proteome (derived from the aPleWal.anno.v2.20220926.gff3 using Trinotate v4.0.1[71] Trinotate_GTF_or_GFF3_annot_prep.pl –annot aPleWal.anno.v2.20220926.gff3 –genome_fa aPleWal1.pri.20220803.fasta –out_prefix aPleWal.anno.v2.20220926.fa

To identify genes putatively absent from the *P. waltl* genome, we queried the LUCA.h5 database associated with OMAmer v2.0.4[75]
omamer search –db LUCA.h5 –query proteome.fa –out output.omamer

We then used OMArk v0.3.0[74] to assess the completeness of the proteome and identify putative absent genes. Splice files were produced for both species linking all proteins isoforms derived from a single gene.
omark -f output.omamer -d LUCA.h5 -i splice_file.txt -o omark_output

We then adapted the Jupyter notebook (Human_missing_genes.ipynb) found here:[79] to identify putatively absent genes. This code can be found here: (https://github.com/aelewa/Pleurodelesgenome/blob/main/20241014_splice_Pleuro_missing_genes.ipynb) and uses biopython, bedtools, BLAST, pandas, and matplotlib. Once genes were identified we followed up with manual inspection of hits via synteny comparisons the *Xenopus tropicalis* (v10), *P. annecetens* (PAN1.0, GCF_019279795.1), *P. waltl*, and axolotl (AmexG_v6), genomes. To confirm gene losses BLAST searches against the *P. waltl* assembly was performed.

Using the human MHC region on chromosome 6 (coordinates 29,602,238-33,409,896) from the GRCh38.p14 assembly as reference, defined by *GABBR1* and *KIFC1* as boundary genes, we extracted all protein-coding genes from NCBI. Initial MHC locus identification in *P. waltl* and *A. mexicanum* was performed using tblastn (e-value 1e-10, bit score ≥300) with human MHC protein sequences against *P. waltl* and *A. mexicanum* transcriptomes. We then examined six regions of *P. waltl* manual_scaffold_6, including four consecutive sections (between coordinates 117,563,038-1,615,656,077) and two MHC locus regions we had identified (coordinates 38,217,551-118,175,120 and 1,615,652,267-1,777,346,995). These regions were compared to *A. mexicanum* transcriptome. Transcript sequences were extracted from the *P. waltl* annotation file (aPleWal.anno.v2.20220926.gff3) and transcriptome (aPleWal1.pri.transcriptome_mRNA.20220803.fasta), and known transposable elements were removed based on a curated list of repeat elements (https://github.com/egypsci/Pleurodelesgenome/blob/main/TEstems_20220707.list). BLAST databases were created using makeblastdb for both *P. waltl* (aPleWal1.pri.transcriptome_mRNA.20220803.fasta) and *A. mexicanum* (AmexT_v47_dna.fa) transcriptomes. Each region was analyzed using blastn in two comparisons: *P. waltl* vs *P. waltl* and *P. waltl* vs *A. mexicanum*, with default parameters (evalue 1e-10). The sequence matches were filtered and annotated using the species-specific annotation files (aPleWal.anno.v2.20220926.gff3 and AmexT_v47-AmexG_v6.0-DD.gtf) to extract chromosome location, gene ID, name, and coordinates for each match. For sequence conservation assessment between *P. waltl* genes, matches with bit scores ≥5000 were considered significant, while for *P. waltl* vs *A. mexicanum* comparisons, a threshold of ≥300 was applied. Key MHC genes including *GABBR1*, *KIFC1*, *TUBB*, *TRIM39* and *GNL1* were identified and used across genomes to establish syntenic relationships.

### Transposable elements

Among the 33,742 gene models encoding conserved protein coding genes, 14,943 genes were putative transposable elements by virtue of protein homology with a transposable element component.

### Current protein-coding gene count

The exclusion of 14,943 putative transposable elements from the 33,742 conserved protein coding genes left 18,799 conserved protein coding genes which is the current protein-coding gene count for *P. waltl.*

## Intron counting

Gene intron coordinates were retrieved from the *P. waltl* genome annotation gff-file using faidx from SAMtools 1.12 and the complement tool from BEDtools 2.30.0. First, the genome annotation was formatted to gtf with AGAT Toolkit[70] (agat_convert_sp_gff2gtf.pl). Afterwards, the chromosome sizes were calculated with faidx, and the intergenic regions were retrieved from the annotation file. Finally, the intron coordinates were obtained by crossing the intergenic regions with the exon coordinates from the annotation file with the bedtools complement command.

## Repeat element annotation and analysis

### Repeat library assembly

The *de novo* repeat libraries of *A. mexicanum, X. tropicalis, and H. sapiens* genomes were developed via RepeatModeler 2.0.2a, with RECON 1.0.8, RepeatScout 1.0.6, and LTRStruct (LTR_retriever 2.9.0 and LTRharvest from genometools 1.6.2) methodologies for the identification of repeat elements. Repeat elements classified by RepeatModeler as "Unknown" were further processed with the DeepTE[80] algorithm to identify their possible family.

A *de novo* repeat library of each genome was made into a database with RepeatModeler BuildDatabase function. Then, a fasta file with predicted classified repeats and their sequences was generated:

RepeatModeler-2.0.2a/RepeatModeler -database "REF_db" -pa 32 -LTRStruct

DeepTE was executed with a metazoan repeat training dataset to classify "Unknown" repeats:

python DeepTE/DeepTE.py -d deep_temp -o REF_DeepTE -sp M -m_dir Metazoans_model
-i repeatFamiliesUnknowns.fa

As a final step, the obtained *de novo* library was combined with the Dfam 3.3 database consensus:

RepeatMasker/famdb.py -i RepeatMasker/Libraries/Dfam.h5 families –ancestors -f fasta_name "organism name" > Dfam.fasta
cat Dfam.fasta db-families.fa > repeatFamilies.fasta

### Masking of transposable elements

To identify the location of the predicted repeats, the combined Dfam and *de novo* library consensus were mapped to the whole genome using RepeatMasker 4.1.2-p1 (Smit, A., Hubley, R., Green, P. RepeatMasker Open-4.0 [WWW Document]. (2013)). with RMBlast 2.11.0 as a search engine. This generated a gff-annotation file with repeat elements position across the genome:

RepeatMasker -pa 32 -gff -xsmall -dir repeat_masker/ -e rmblast
-libdir RepeatMasker/Libraries -lib repeatFamilies.fasta genome.fa

### Classification procedure of repeat elements

To assess repeat composition, a custom bash script was written to process files in an automatic manner. Classification of repeats was based on[81] and was expanded with further information on unclassified ancient repeats from Repbase.[82] The categorization of other non- transposable elements was based on the Dfam classification. Each element was assigned to a class, a superfamily, a family, and a category, as shown in the below table.

| Class | Subclass | Superfamily | Type |
|---|---|---|---|
| ClassI | nLTR | R2 | LINE |
| ClassI | nLTR | MIR/MIR-core | SINE |
| ClassI | LTR | Ty3/Gypsy | LTR |
| ClassII | Cryptons | CryptonA/Crypton-A | DNA |
| ClassII | DD (E/D) transposons | hAT | DNA |
| Other | Simple_repeats | NA | non-TE |

The generated repeat IDs and their corresponding names from RepeatModeler and DeepTE were combined into a dictionary with added classifications. Files were parsed with GNU Awk, where each string was split to retrieve a family name. This name was then compared with the file with classification structure. In the case of repeats that aligned with the Dfam database consensus, no IDs were generated. Thus, they were processed separately and repeat classification was retrieved from Dfam classification using RepeatMasker inner function famdb.py. The result was a repeat annotation gtf-file with coordinates, the names of repeat elements, and full classification. This annotation was subsequently used to quantify repeat family contributions to the genome size of *P. waltl*, *A. mexicanum*, *X. tropicalis*, and *H. sapiens*.

### Genomic location of the annotated repeats

To identify the location of repeat elements within introns, exons and intergenic regions of the *P. waltl* genome, their coordinates were intersected with bed-files generated in the 'Intron counting' section (see above) via BEDtools intersect tool.

To test if the hAT elements are over- or underrepresented in the intergenic, intronic or exonic regions of the *P. waltl* genome, binomial test was performed.

Region: intergenic
Observed Count: 1797804528.77
Expected Count: 1573570703.52
P-value: 1.457494e-321
Adjusted P-value: 1.467375e-321
A higher observed count suggests that hATs are over-represented in the intergenic region.
Region: intronic
Observed Count: 515112997.95
Expected Count: 682651996.38
P-value: 1.417968e-321
Adjusted P-value: 2.124482e-321
The lower observed count suggests that the longer *P. waltl* intronic length is not due to the hAT elements.
Region: exonic
Observed Count: 462814.91
Expected Count: 57851864.10
P-value: 4.891250e-322
Adjusted P-value: 1.467375e-321
The significantly lower observed exonic count suggests that exons are conserved and are not mutated due to the insertions of hAT elements.

### P. waltl transcriptome analysis

### Analysis of hAT family

For expression analysis, transposons from the hAT family as well as domesticated hATs genes (ZBED) were used. Due to the existence of more than 4 million entries in our hAT annotation file, hAT transposons' coordinates were split into different files based on their chromosome location and further analysis was done separately. Raw RNA-seq data was only used for counting hAT transcripts. The raw counts of ZBED genes were available.[11]

### Iso-seq data analysis

Iso-Seq data was mapped to *P. waltl* genome with minimap2 2.24 with following parameters:

```
minimap2 -I200G -t24 -axsplice:hq --secondary=no -uf -G5m $genome_file $isoseq_file
```

The resulting sam file was transformed into a bam format, sorted by coordinates and indexed creating a csi-file to accelerate further transcript counting:

```
samtools view -@24 -b isoseq.pWaltl.mapped.sam | samtools sort -@24 -o isoseq.pWaltl.sorted.bam - && samtools index -c -@24 isoseq.pWaltl.sorted.bam
```

The number of reads for hAT transposons and ZBED genes was acquired with featureCounts from subread 2.0.1 package:

```
featureCounts -t similarity -g gene_id -L -a annotation.gtf -o $outfile $bam_file
```

**❁ CellPress**

**Cell Genomics**

### RNA-seq data analysis

STAR 2.7.9a package was used to index the *P. waltl* genome and then map RNA-seq transcripts to it:

    STAR –runThreadN 24 –runMode genomeGenerate –genomeDir ./indexed_genome
    –genomeFastaFiles $genome_file –sjdbGTFfile $full_annotation.gtf –sjdbOverhang
    124 –limitGenomeGenerateRAM 55000000000

    Mapping was performed using parameters previously reported[11]

    STAR –genomeLoad LoadAndRemove –genomeDir $genome_dir –runThreadN 24
    –readFilesIn rnaseq1_1.fastq rnaseq1_2.fastq –outFileNamePrefix "rnaseq1_tr_"
    –limitGenomeGenerateRAM 300000000000 –runDirPerm All_RWX –outFilterMultimapNmax 1000 –outFilterMismatchNoverLmax
0.05 –alignIntronMax 1 –alignIntronMin 2
    –scoreDelOpen -10000 –scoreInsOpen -10000 –alignEndsType EndToEnd
    –limitOutSAMoneReadBytes 100000000000

    Sam-file transformations were performed as described for the Iso-Seq analysis. Then transcripts were counted: featureCounts -p
-T 10 -t similarity -g gene_id -a $hAT.gtf -o $outfile $bam

### Data processing

Differential expression of RNA-seq data was processed with DESeq2 R package. DESeq2's median of ratios was used to normalise raw counts to measure up- or downregulation of transposons and genes. Transcriptional count mean-dispersion relationship was performed with parametric fit or local regression fit (transposons). For hATs, analysis was only performed on predicted repeats that were longer than 2000 bp. Samples from 0 days post amputation (dpa) and 3 dpa or 7 dpa were compared pairwise to determine significantly differentially expressed genes/transposons. Genes or repeats were filtered by adjusted p-value (<0.1) for visualisation. After regularised log transformation and data centring Z-scores were calculated and visualised as a heatmap applying pheatmap package. All other plots were generated using package "ggplot2" from "tidyverse".

### Reverse transcription quantitative real-time PCR

RNA from limb or blastema tissues from 9-month-old newts (6-7cm snout-to-cloaca) was isolated using standard Trizol method (n = 4). cDNA was prepared using Superscript-IV kit. qPCR was carried out using iQ SYBR Green supermix (Bio-rad, Hercules, CA) on a Chromo 4 instrument running Opticon 3 software (Bio-rad). Gene expression relative to *Ef1*a was calculated using the Livak $2^{-\Delta\Delta Cq}$ method. The primers used were *Ef1α* forward: CTCCACCGAACCTCCTTACA, *Ef1α* reverse: CCAGCCTTTAAACCAGGTCA, *Zbed6.1* forward: GCTGCTGATGATGACACTGG, *Zbed6.1* reverse: AGGACTGAGCTCTAGGGTGA, *Zbed1.1* forward: TGCAGTGGATGGATGTCTGT, *Zbed1.1* reverse: GCTCATCTGGCATCTGAAGC, *Charlie.76* forward: CCTGAGGAAACCGATCATGTCT, *Charlie.76* reverse: GCCAGCCAAAATTCTCCCAA.

### circRNA characterisation

Firstly, reads originating from 18 total RNA-seq libraries were aligned to the newly assembled *P. waltl* genome using bwa v0.7.17. Default parameters were then used to run all three circRNA detection programs; CIRCexplorer2[40], Find_Circ2[41] and CIRI2.[42] All the output count files were further processed in R with circRNAprofiler v1.14.0[83] as described in the manual.

To find inverted repeat elements in flanking introns, Bedtools intersect was used to find those overlapping at least 50% with introns. Followed by Bedtools closest to find repeat elements closest to back-spliced junctions. Inverted repeat elements were defined by the closest up- and downstream repeats on opposite strands.

Given a higher genomic abundance of repeat elements in the *P. waltl* genome, we test whether the association of hATs, LTRs and nMITEs with the formation of circRNAs is significant. We performed a binomial test as follows:

    hATs 11.4%
    LTRs 26.40%
    nMITES 3.22%
    Total length nMITES 654019872 nt
    Total length LTR 4463436748 nt
    Total length hAT 2314074564 nt
    Element: hATs
    Observed Count: 2297
    Expected Count: 1117.086
    Expected Proportion: 0.114
    P-value: 1.174142e-245
    Adjusted P-value: 0.000000e+00
    Element: LTRs
    Observed Count: 4917
    Expected Count: 2586.936
    Expected Proportion: 0.264
    P-value: 0.000000e+00

Adjusted P-value: 0.000000e+00
Element: nMITEs
Observed Count: 1915
Expected Count: 315.527
Expected Proportion: 0.032
P-value: 0.000000e+00
Adjusted P-value: 0.000000e+00

Based on above results we reject the null hypothesis as the observed count of circRNAs are significantly over-represented compared to the expected count.

For comparative analysis, the 18 libraries were grouped into the following five categories: Developing animals (St. 25-29 and 33-35, denoted as "Developing larvae"), Adult tissues (brain, eyes, heart, liver and lung), Forelimb (uninjured adult limb), Limb Stump (adult Day 0) and Regenerating limb (adult Day 3 and Day 7 dpa). To generate PCA plots log-normalised counts per million were calculated using edge-R v3.42.4. Pairwise analysis of differential circRNA expression was performed with circRNAprofiler using getDeseqRes() function adapted from DESeq2.

## miRNA identification and genomic distribution

We merged and modified established tools to identify genomic loci containing miRNA precursor sites. We took two independent approach of miRNA predictions. First, using miRDeep2[52] that relies on small RNA-seq therefore identifying likely expressed miRNA candidates. Second, MiRmachine prediction that utilises a covariance model, which is based on manually curated microRNA database, MirGeneDB.[53]

Using the small RNA-seq libraries as input, previously annotated salamander miRNAs[11], and miRBase annotations for, *X. tropicalis*, miRNAs were predicted by the miRDeep2 programme.[52] First, we parsed all 10 previously published small RNA-seq datasets from various *P. waltl* tissues[11] through miRTrace (v1.0.1)[51] with default settings to filter miRNA reads as a quality control step, while clipping the adaptors (Data S6). This results in a file of filtered and collapsed reads.

mirtrace qc –species xtr –adapter TGGAATTCTCGGGTGCCAAGG
SRR6001100_small_RNA-seq_of_P._waltl_late_embryo.fastq.gz
SRR6001101_small_RNA-seq_of_P._waltl_larvae.fastq.gz
SRR6001102_small_RNA-seq_of_P._waltl_oocytes.fastq.gz
SRR6001104_small_RNA-seq_of_adult_P._waltl_heart.fastq.gz
SRR6001105_small_RNA-seq_of_adult_P._waltl_lung.fastq.gz
SRR6001107_small_RNA-seq_of_adult_P._waltl_eyes.fastq.
SRR6001127_small_RNA-seq_of_adult_P._waltl_brain.fastq.gz
SRR6001128_small_RNA-seq_of_adult_P._waltl_regenerating_forelimb_7dpa.fastq.gz
SRR6001130_small_RNA-seq_of_adult_P._waltl_liver.fastq.gz
SRR6001103_small_RNA-seq_of_adult_P._waltl_regenerating_forelimb_3dpa.fastq.gz –write-fasta

The second step involved generating a new -64-bit index of the genome using bowtie (v1.3.1). We built a bowtie 64-bit index to accommodate the giant size of the genome without splitting large chromosomes and mapped the filtered, miRNA-enriched reads to the genome with an overall alignment rate of 75.50%.

bowtie-1.3.1-linux-x86_64/bowtie-build –large-index aPleWal1.scaffolded.masked.fa.gz PleWal1_bowtie2

The third step consisted in merging both, collapsed reads (output of miRTrace) and -64bit index from above with mapper.pl module (mirdeep2-v0.1.3).[52]

mapper.pl Config_fa.txt -d -c -i -j -l 18 -m -o 18 -p PleWal1_bowtie2 -t reads_vs_genome.arf -v -n

As final step, we ran miRDeep2.pl (mirdeep2-v0.1.3).

Given the numerous repeat regions in the genome, we modified the script and set the mismatches to zero when mapping reads against the excised precursors. This is done by changing the script in prepare_signature.pl from

system ("bowtie -f -v $read_align_edit_distance -a –best –strata –norc $dir/precursors.ebwt $file_reads $dir/reads_vs_precursors.bwt 2> /dev/null");

to

system ("bowtie -f -v 0 -a –best –strata –norc $dir/precursors.ebwt $file_reads $dir/reads_vs_precursors.bwt 2> /dev/null"); and then running the below code.

miRDeep2.pl all_like.fasta aPleWal1.scaffolded.masked.fa reads_vs_genome.arf newt_MAT3.fasta output_xenfasta newt_PRE3.fasta 2>mirdeep_mismatchnew.log

all_like.fasta – output of miRTrace from all ten small RNA-seq datasets
aPleWal1.scaffolded.masked.fa – newly sequenced genome
reads_vs_genome.arf – output of mapper.pl module
newt_MAT3.fasta and newt_PRE3.fasta – previously annotated mature and precursor sequences
output_xenfasta – *Xenopus tropicalis* (xtr) mature seq from miRBase as related species.

To minimise inclusion of false positive predictions all miRNAs predicted by miRDeep2 (3046 miRNA candidates) were filtered for those with a miRDeep2 score of 3.9 or greater (855 miRNA candidates), which corresponded to a high signal to noise ratio (Data S7). In a repeat element enriched genome, such as *P. waltl,* multimapping may cause an inflated number of miRNA candidates. However, excluding multimapping reads risks underestimating genuine miRNA read counts as multiple miRNA precursors could produce the same mature miRNA. Nevertheless, miRDeep2 accounts for multimapping by discarding reads which have more than five alignments. Although this strategy limits the contribution of multimapping, we still explored this contribution by examining only uniquely mapping small RNA-seq reads. Briefly, we created a new bowtie index based on all 855 candidate precursor sequences and used bowtie to align all small RNA-seq reads excluding reads with more than one alignment. This way we identified the large majority, >85.9%, of miRNA candidates (735/855) that had at least one read mapped uniquely to their precursors compared to >78.5% that had five or more uniquely mapped reads (672/855).

bowtie -v 0 -m 1 –al output -p 10 -f miRindex all_like.fasta

For each of the four species-specific miRNA groups we re-ran this same analysis with an index based on all precursors in each group (Table S15).

To unveil the genomic organisation of all the miRNAs in the intergenic, intronic and exonic regions of the genome, introns were added to the gtf file using AGAT.

agat_sp_add_introns.pl –gff unsplit.gtf -o unsplit_vidintrons.sort.gtf

Subsequently, Bedtools (v2.30.0) was used to intersect with the gtf file to reveal the distribution of miRNAs in the *P. waltl* genome. To generate a heatmap of miRNA expression, the quantifier module of miRDeep2 was used and normalised reads were plotted.

quantifier.pl -p all_precursor.fa -m all_mature.fa -r all_like.fasta -W

To test if the embedding of annotated miRs in the LTR repeat element is by chance due to LTR being genomically abundant in *P. waltl* genome, we performed a binomial test.

```
import scipy.stats as stats
genome_size = 20300780447
te_coverage = 4463436748
total_new_mirnas = 855
overlapping_mirnas = 214
Probability of overlap by chance
p = te_coverage / genome_size
print(p)
0.22
p_value = stats.binom_test(overlapping_mirnas, total_new_mirnas, p, alternative='greater')
print(f"P-value: {p_value}")
P-value: 0.019
Are novel annotated miRs in LTR Repeat elements by chance?
total_new_mirnas = 60
overlapping_mirnas = 42
P-value: 2.82e-15
```

Based on the significant results, we reject the null hypothesis. This suggests that embedding of miRs in the LTR repeat element is not by chance. However, we do not know what mechanism drives LTR to be the source of miRs in the newt.

Chromosome location of All- and ESCC-miRNAs were taken from the Table S9 to plot Figure 5E. The miRNAs were plot to chromosomes using R package, chromPlot (v1.22.0).[84]

Genomic distribution pie-charts (Figure 5C) and bar plots (Figure 5D) were created using matplotlib package in Python.

MirMachine v0.2011.2022 was used with default parameters.

Parameters: MirMachine.py –node Amphibia –species aPleWal1 –genome aPleWal1.pri.20220803.fasta –model deutero –cpu 32

Heatmap was created in R using reshape2 & ggplot2 packages based on MirMachine predictions.[53]

### MAFFT alignment and tree map

The precursor sequences of 43 ESCC-miRNAs were uploaded on MAFFT server (version 7) (https://mafft.cbrc.jp/alignment/server/). The alignment was visualised on JalView (v2.11.2.4) by highlighting percentage identity. Unrooted tree map was generated by uploading the MAFFT aligned 43 ESCC-miRNA precursors using Interactive Tree Of Life (iTOL) v5.

## QUANTIFICATION AND STATISTICAL ANALYSIS

The quantitative and statistical analyses are described in the relevant sections of the method details or in the figure legends.

