## [Document S2. Transparent peer review records for Brown et al. (TPR_Simon.pdf) · Cell Genomics]

Chromosome-scale genome assembly reveals how repeat elements shape non-coding RNA landscapes active during newt limb regeneration

Thomas Brown, Ketan Mishra, Ahmed Elewa, Svetlana Iarovenko, Elaiyaraja Subramanian, Alberto Joven Arous, Andreas Petzold, Bastian Fromm, Marc R. Friedländer, Lennart Rikk, Miyuki Suzuki, Ken-ichi T. Suzuki, Toshinori Hayashi, Atsushi Toyoda, Catarina Oliveira, Ekaterina Osipova, Nicholas D. Leigh, Maximina H. Yun
András Simon

Summary

Initial submission: Received : Jul 02, 2024

Scientific editor: Sara Rohban

First round of review: Number of reviewers: 2
Revision invited : Aug 13, 2024
Revision received : Dec 04, 2024

Second round of review: Number of reviewers: 1
Accepted : Jan 03, 2025

Data freely available: YES

Code freely available: YES

This transparent peer review record is not systematically proofread, type-set, or edited. Special characters, formatting, and equations may fail to render properly. Standard procedural text within the editor's letters has been deleted for the sake of brevity, but all official correspondence specific to the manuscript has been preserved.

Referees' reports, first round of review

Reviewer #1:

I have reviewed the submission by Brown et al., which has existed as a preprint for ~2 years and permeated widely through the field at this point. Overall, it is a competent and crucial contribution to the field of genomics, and there are relatively few obstacles to its publication. Following is a list of relatively minor comments for the AE and the authors to consider:

1. While it's unlikely the authors will want to completely re-assemble the genome, it's important to note that hifiasm now has a Hi-C module that provides for integrated phasing and purging during the initial assembly, obviating tools like `purge_dups` and possibly `deep_consensus`. This might provide substantial increases in the "3 Cs" and would be an excellent late addition, if possible. Similarly, `compleasm` is now a much higher-quality tool for assessing completeness, and may provide better scores than BUSCO. I see that the authors have added in a note about a haplotype-resolved assembly using hifiasm-0.1.6.0, but I don't see many details about this new version.
2. The graphics have not been updated substantially and leave a bit to be desired. It would be very interesting to see a Circos plot comparing to *Ambystoma*, and something like Figure 2 from Smith et al. 2019 (a dot plot-ideogram) to show short- and long-term synteny mapping and chromosomal divergence. This would be a good addition to or replacement for the dot plot in Figure 3, which is kind of hard to make out for the smaller segments.
3. While I realize that comparative evolutionary genomics is not the primary focus of this MS, and the authors have limited their discussion to comparable chromosome-scale assemblies - a reasonable decision - there is still a substantial literature on salamander genomics they do not touch on. Rachel Lockridge Mueller's lab at CSU has done a substantial amount of work on TE content and its contribution to genome-size changes in salamanders:
<https://academic.oup.com/gbe/article/4/2/168/557286>
<https://www.ncbi.nlm.nih.gov/pmc/articles/PMC2435017/>
<https://academic.oup.com/gbe/article/6/7/1818/553594>

Additional data are available in the complete but unscaffolded *_Desmognathus_* genome that was just released: <https://www.biorxiv.org/content/10.1101/2024.04.30.591895v1>

Given the amount of focus here on TE content variation (e.g., Figure 2), briefly mentioning some of this preliminary data from other salamanders might be appropriate.

4. The authors seemingly downplay their genomic annotations - it would be good to get a bit more context and description here. How do the CDS/ORF/GO terms etc. compare to other assemblies? Are any key interesting genes missing in comparison to *Ambystoma* or other tetrapods? Does *Pleurodeles* have PRDM9, for instance? I assume not, but it would be interesting to get some insights here. Do they have the same interesting regeneration and MHC complexes as Schloissnig reported in axolotls? Are there any surprises in gene content?

These comments are all minor and within the discretion of the authors and the AE to address as they see fit.

Reviewer #2:

It is exciting to see another gigantic genome assembled, and the potential to unlock molecular mechanisms underlying both regeneration and genome expansion is a real strength of this manuscript. My primary comments are about framing and contextualizing the results, which I hope will allow the authors to revise their manuscript for greater clarity and impact.

First, I would suggest the authors give the readers more information about how the specific results they report compare to the genome-wide expectation (or transcriptome-wide) expectation in order to help readers understand the significance of the patterns reported. For example:

P 5 188-198 reporting results of differential expression of TEs during regeneration and across tissues. Could the authors give the readers some idea (based on the literature, or their own analyses) of whether one would expect TE expression to differ across tissues or during development? In other words, is this the pattern you'd predict based on our null expectation emerging from TE biology? Or is this unusual?

P 4 lines 175-177 How do these percentages of hAT element distribution compare to the genome itself? In other words, is 76% of the genome intergenic, in which case the TE insertion matches the null expectation? Or are the TEs over- or underrepresented in certain genomic regions?

P 6 251-52 Similar question here - how many miRNAs would you expect to be within 100 bp of a repeat element just by chance, given the overall genomic TE landscape?

P 8 351-53 are hATs, LTRs, and MITEs associated with circRNAs at higher levels than you would predict based on their genomic abundance? Along these same lines, is there something specific about the replicative mechanism or insertion profile of hATs (or the others) that you posit might explain their association with circRNAs? The more you can connect these dots for the reader, the more the significance of the work will become clear.

Second, I suggest the authors revise the overall framing to better set up the significance of the results. The authors frame the manuscript around large genome size and circRNA discovery, but I think the framing could be improved by more explicit connection between genome expansion (via TE accumulation) and circRNA formation. What processes are expected to be accentuated in large genomes (ie TE proliferation, increased intron length, etc.) and by what mechanism would those be predicted to impact circRNA formation? Does having a huge genome change any expectations for circRNA conservation or function? This would help the reader to understand the significance of the results better and improve the impact of the manuscript. This comes through in the Discussion (e.g. line 345), but I think it should be clarified and set up upfront as the framing for the work.

The mention that *P. waltl* has unique regeneration mechanisms (p 9 381-84) here also raises for me the invitation to expand a little bit more. I would suggest the authors spend a little more time discussing the differences between regenerating limbs across newts and salamanders and whether there is any potential connection they would posit between whatever is unique to *P. waltl* and their circRNA profile and/or miRNA profile.

I would recommend saying a bit more about the function of circRNAs the first time they are introduced in the manuscript for unfamiliar readers.

Third, I would encourage the authors to be a little conservative about the interpretation connecting expression to function in regeneration (e.g. P 8 line 340). They are careful in their wording (i.e. Hence, hATs, members of Class II repeat element, may represent previously unrecognized molecular components of vertebrate limb regeneration.) I agree that this statement is consistent with the results, but it also strikes me that it is hard to disentangle cause and effect here - some repeat elements are likely transcribed as a by-product of open chromatin and active expression of nearby transcripts. I would encourage the authors to be upfront about both of these possibilities, but I agree the result is exciting and sets a rich stage for further investigation.

Authors' response to the first round of review

Reviewer#1

I have reviewed the submission by Brown et al., which has existed as a preprint for ~2 years and permeated widely through the field at this point. Overall, it is a competent and crucial contribution to the field of genomics, and there are relatively few obstacles to its publication. Following is a list of relatively minor comments for the AE and the authors to consider.

Thank you for the positive and encouraging overall assessment.

1. While it's unlikely the authors will want to completely re-assemble the genome, it's important to note that hifiasm now has a Hi-C module that provides for integrated phasing and purging during the initial assembly, obviating tools like purge_dups and possibly deep_consensus. This might provide substantial increases in the "3 Cs" and would be an excellent late addition, if possible. Similarly, compleasm is now a much higher-quality tool for assessing completeness, and may provide better scores than BUSCO. I see that the authors have added in a note about a haplotype-resolved assembly using hifiasm-0.1.6.0, but I don't see many details about this new version.

Thank you for the suggestions. Given that our assembly has already been widely used in the field, as the reviewer points out above, we retained the original assembly in the revised version. As an important addition and improvement, we included the statistics from the two haplotypephased assemblies in the results section alongside those of the primary assembly (line 101-103, S Table 1). BUSCO analysis using the new miniprot mode corroborated the outcome of our previous analysis using compleasm. As such, all three assemblies displayed a high level and comparable contiguity but the primary assembly has the highest BUSCO scores and contiguity. However the phased assemblies were particularly useful when identifying the large inversions identified in chromosomes 2 and 5, which we highlighted in the results section (line 126-131).

2. The graphics have not been updated substantially and leave a bit to be desired. It would be very interesting to see a Circos plot comparing to *Ambystoma*, and something like Figure 2 from Smith et al. 2019 (a dot plot-ideogram) to show short- and long-term synteny mapping and chromosomal divergence. This would be a good addition to or replacement for the dot plot in Figure 3, which is kind of hard to make out for the smaller segments.

As requested by the reviewer we made a new figure 2 with new data analyses and new type of representation of the data in the previous version. The old figure has been moved to the supplementaries (S. Fig. 6D,E). Specifically we show the following:

(a) Oxford plot depicting the locations of Chordate Linkage Groups (CLGs). A total of 14,062 best-reciprocal-hit orthologs were identified between the *P. waltl* chromosomes and 17 ancestral CLGs. The dense rectangular blocks of dots represent units of deeply conserved synteny. The Fisher's exact test (FET) is used to calculate the significance of interactions between each scaffold and Linkage Group. Dots are coloured by CLGs, with solid dots indicating the FET p -value ≤ 0.05 and translucent dots depicting the FET p -value > 0.05 .

(b) Ribbon plot showing syntenic blocks of the *newt*, *axolotl* and *gar* genomes. Bestreciprocal-hit orthologs are connected by ribbons between the *P. waltl*, *A. mexicanum* and *L. oculatus* genomes and coloured based on their identification as proteins from the 17 CLGs as in (a).

Regions of the *newt*, *axolotl* and *gar* genomes identified pairwise as containing co-localised blocks of genes with the CLGs between *axolotl* and *newt* (c), *gar* and *newt* (d). The size of each circle corresponds to the number of genes in each identified syntenic block. Circles are coloured by CLGs. Individual numbers are also available in Supplementary Tables 2-6.

(e) Microsynteny analysis of the *Fgf5* locus between *gar*, lungfish, frog, *newt* and *axolotl*.

The *newt* locus has experienced various disruptions including a 58 Mb gap between *Prdm8* and *Cfap299*, as well as an inversion of order between *Paqr3* and *Prdm8*. Arrow represents the relative position and direction of genes on chromosomes, retrogenes are marked by white rectangles and filled rectangles imply a different chromosomal position.

3. While I realize that comparative evolutionary genomics is not the primary focus of this MS, and the authors have limited their discussion to comparable chromosome-scale assemblies - a reasonable decision - there is still a substantial literature on salamander genomics they do not touch on. Rachel Lockridge Mueller's lab at CSU has done a substantial amount of work on TE content and its contribution to genome-size changes in salamanders:

<https://academic.oup.com/gbe/article/4/2/168/557286>

<https://www.ncbi.nlm.nih.gov/pmc/articles/PMC2435017/>

<https://academic.oup.com/gbe/article/6/7/1818/553594>

Additional data are available in the complete but unscaffolded *_Desmognathus_* genome that was just released: <https://www.biorxiv.org/content/10.1101/2024.04.30.591895v1>

Given the amount of focus here on TE content variation (e.g., Figure 2), briefly mentioning some of this preliminary data from other salamanders might be appropriate.

We updated our analyses on TEs (line 250-251, S Fig. 15) and we also refer to prior contributions in the introduction and discussion as suggested by the referee (lines 62-64 and 404-406).

4. The authors seemingly downplay their genomic annotations - it would be good to get a bit more context and description here. How do the CDS/ORF/GO terms etc. compare to other assemblies? Are any key interesting genes missing in comparison to *Ambystoma* or other tetrapods? Does *Pleurodeles* have PRDM9, for instance? I assume not, but it would be interesting to get some insights here. Do they have the same interesting regeneration and MHC complexes as Schloissnig reported in axolotls? Are there any surprises in gene content?

We agree that the possibilities with the genomic annotation were not sufficiently highlighted in the previous version. In the present version we have made substantial improvement and as outlined above we have made a new Fig. 2. As requested by the reviewer we investigated specifically the case of PRDM9 and could conclude that the gene was missing (lines 179-180).

The independent annotation made by NCBI also confirms this conclusion

*(<https://www.ncbi.nlm.nih.gov/datasets/taxonomy/8319/>). We again thank the reviewer for the suggestion and as requested, we analysed MHC locus and found rearrangements that we report in the revised version (lines 190-199 and S. Fig. 11). The bimodal distribution of MHC locus in *P. waltl* and its implication on immune system and regeneration will be investigated in future.*

These comments are all minor and within the discretion of the authors and the AE to address as they see fit.

Thank you, we believe we addressed each comment with new analyses and clarifications.

Reviewer#2

It is exciting to see another gigantic genome assembled, and the potential to unlock molecular mechanisms underlying both regeneration and genome expansion is a real strength of this manuscript. My primary comments are about framing and contextualizing the results, which I hope will allow the authors to revise their manuscript for greater clarity and impact.

We thank the reviewer for their request to improve the framing and contextualization of the paper which we did according to the suggestions.

P 5 188-198 reporting results of differential expression of TEs during regeneration and across tissues. Could the authors give the readers some idea (based on the literature, or their own analyses) of whether one would expect TE expression to differ across tissues or during development? In other words, is this the pattern you'd predict based on our null expectation emerging from TE biology? Or is this unusual?

*To test whether there is specific pattern of TE expression indicative of their selective regulation during regeneration we analysed their expression. We found that unlike to axolotl in which LINE1 elements have been shown to be upregulated as an early response to limb amputation, this is not the case for *Pleurodeles* (lines 248-251 and new S Fig. 15), further highlighting divergent regeneration mechanisms acting in closely related species (see also Sandoval-Guzman, 2014). Given that we observe that DNA TE elements are dominating in *Pleurodeles* we focused on their expression pattern during limb regeneration*

(new S Fig. 15 A and B). As suggested, we also performed binominal tests which reject the null hypothesis, further corroborating our results of specific expression of TEs (lines 873-902).

P 4 lines 175-177 How do these percentages of hAT element distribution compare to the genome itself? In other words, is 76% of the genome intergenic, in which case the TE insertion matches the null expectation? Or are the TEs over- or underrepresented in certain genomic regions?

Upon request, we performed a statistical analysis on the distribution of the hAT elements and show in the revised version that they do not match the null hypothesis. hAT elements contribute significantly more to the expansion of the intergenic region ($p_{adj} < 1.47e-321$, lines 873-902).

P 6 251-52 Similar question here - how many miRNAs would you expect to be within 100 bp of a repeat element just by chance, given the overall genomic TE landscape?

As asked by the reviewer we performed the statistical analysis (lines 1101-1130). LTR repeat elements are not the major contributor to the genome yet contained most of the reported miRNAs. Although the molecular underpinning of this distribution pattern is not known at present, our findings support the hypothesis that novel miRNAs originate from inverted repeats with the potential to form stem-loop structures that subsequently might evolve into functional miRNA genes (Smalheiser, 2005; Cho, 2018).

P 8 351-53 are hATs, LTRs, and MITEs associated with circRNAs at higher levels that you would predict based on their genomic abundance? Along these same lines, is there something specific about the replicative mechanism or insertion profile of hATs (or the others) that you posit might explain their association with circRNAs? The more you can connect these dots for the reader, the more the significance of the work will become clear.

As asked by the reviewer we performed the statistical analysis (lines 973-1005). Although the data does not provide a mechanistic explanation for replicative mechanism, the null hypothesis could be rejected as we found preferential association to hATs, LTRs and nMITES (Fig. 4d). We also discuss in more detail the significance of these data both in the Introduction (lines 80-82) and in the Discussion (lines 429-435).

Second, I suggest the authors revise the overall framing to better set up the significance of the results. The authors frame the manuscript around large genome size and circRNA discovery, but I think the framing could be improved by more explicit connection between genome expansion (via TE accumulation) and circRNA formation. What processes are expected to be accentuated in large genomes (ie TE proliferation, increased intron length, etc.) and by what mechanism would those be predicted to impact circRNA formation? Does having a huge genome change any expectations for circRNA conservation or function? This would help the reader to understand the significance of the results better and improve the impact of the manuscript. This comes through in the Discussion (e.g. line 345), but I think it should be clarified and set up upfront as the framing for the work.

According to the reviewer's suggestion we revised the Introduction for having a clearer overall framing of the studies (line 80). "The formation of circRNAs is facilitated by long introns and flanking repeat elements, both of which are characteristics of the salamander genome". Given that the circRNA is an emerging field, the specific mechanisms by which circRNAs are forming remains unknown. However, as noted above the null hypothesis could be rejected as we found preferential association to hATs, LTRs and nMITES (lines 973-1005). As large vertebrate genomes have not been investigated for the circRNAs, we had no prior expectations. We showed that salamander circRNAs extend the hypothesis (Gruhl, 2021) of repeat element involvement and have clarified it further for the readers in Discussion (lines 429-433).

The mention that *P. waltl* has unique regeneration mechanisms (p 9 381-84) here also raises for me the invitation to expand a little bit more. I would suggest the authors spend a little more time discussing the differences between regenerating limbs across newts and salamanders and whether there is any potential connection they would posit between whatever is unique to *P. waltl* and their circRNA profile and/or miRNA profile.

As suggested by the reviewer we expanded the paragraph which describes the comparisons between newt and axolotl (lines 474-477). An in-depth comparison of the miRNA and circRNA annotations between the species is not possible at present because those have not been done for the axolotl.

I would recommend saying a bit more about the function of circRNAs the first time they are introduced in the manuscript for unfamiliar readers.

As recommended by the reviewer, we introduced the function of circRNAs in the Introduction (line 78-80).

Third, I would encourage the authors to be a little conservative about the interpretation connecting expression to function in regeneration (e.g. P 8 line 340). They are careful in their wording (i.e. Hence, hATs, members of Class II repeat element, may represent previously unrecognized molecular components of vertebrate limb regeneration.) I agree that this statement is consistent with the results, but it also strikes me that it is hard to disentangle cause and effect here - some repeat elements are likely transcribed as a by-product of open chromatin and active expression of nearby transcripts. I would encourage the authors to be upfront about both of these possibilities, but I agree the result is exciting and sets a rich stage for further investigation.

As suggested, we stated the possibilities described by the reviewer upfront in the Discussion (lines 413-415).

Referees' report, second round of review

Reviewer #1:

The authors have responded to my queries and those of the other reviewer in detail. The new graphics are excellent, and the discussion of coding elements and RNA structures is detailed and very helpful. I have no further comments.